# ParamMute: Suppressing Knowledge-Critical FFNs for Faithful Retrieval-Augmented Generation

**Pengcheng Huang[1], Zhenghao Liu[1]\*, Yukun Yan[2], Haiyan Zhao[2], Xiaoyuan Yi[3],**
**Hao Chen[2], Zhiyuan Liu[2], Maosong Sun[2], Tong Xiao[1], Ge Yu[1], Chenyan Xiong[4]**

[1]School of Computer Science and Engineering, Northeastern University, China
[2]Department of Computer Science and Technology, Institute for AI, Tsinghua University, China
[3]Microsoft Research Asia, Beijing, China
[4]Language Technologies Institute, Carnegie Mellon University, United States

## Abstract

Large language models (LLMs) integrated with retrieval-augmented generation (RAG) have improved factuality by grounding outputs in external evidence. However, they remain susceptible to unfaithful generation, where outputs contradict retrieved context despite its relevance and accuracy. Existing approaches aiming to improve faithfulness primarily focus on enhancing the utilization of external context, but often overlook the persistent influence of internal parametric knowledge during generation. In this work, we investigate the internal mechanisms behind unfaithful generation and identify a subset of mid-to-deep feed-forward networks (FFNs) that are disproportionately activated in such cases. Building on this insight, we propose Parametric Knowledge Muting through FFN Suppression (ParamMute), a framework that improves contextual faithfulness by suppressing the activation of unfaithfulness-associated FFNs and calibrating the model toward retrieved knowledge. To evaluate our approach, we introduce CoFaithfulQA, a benchmark specifically designed to evaluate faithfulness in scenarios where internal knowledge conflicts with accurate external evidence. Experimental results show that ParamMute significantly enhances faithfulness across both CoFaithfulQA and the established ConFiQA benchmark, achieving substantial reductions in reliance on parametric memory. These findings underscore the importance of mitigating internal knowledge dominance and provide a new direction for improving LLM trustworthiness in RAG. All codes are available at https://github.com/OpenBMB/ParamMute.

## 1 Introduction

Large language models (LLMs), such as GPT-4 [49] and LLaMA [59], have demonstrated exceptional performance across a wide range of natural language processing tasks [12, 38, 64, 77, 78]. Nonetheless, they are known to suffer from hallucinations, frequently generating factually incorrect or fabricated information [14, 25, 42]. To address this, retrieval-augmented generation (RAG) has emerged as a promising paradigm, grounding model outputs in external evidence to improve factual accuracy [35, 76]. Despite these advancements, recent studies [1, 71] have identified a persistent and subtle challenge: LLMs may still produce unfaithful responses that contradict or disregard external evidence even when this evidence is accurate and highly relevant [43, 67]. Such unfaithful generation can significantly undermine the reliability of RAG systems [26].

Recent approaches primarily seek to improve contextual faithfulness by enhancing the model's ability to incorporate external evidence—either through advanced prompting strategies [27, 79] or context-

---

\* indicates corresponding author.

39th Conference on Neural Information Processing Systems (NeurIPS 2025).

aware decoding techniques [1, 23]. However, these externally focused methods often overlook the role of internal knowledge in undermining generation faithfulness. Motivated by this gap, we turn our attention to examining how parametric knowledge influences the generation process. Specifically, we focus on the feed-forward networks (FFNs) within Transformer-based LLMs, which are widely recognized as key repositories of memorized knowledge [10, 20]. Indeed, our pilot study reveals that when a specific subset of mid-to-deep FFN layers exhibits excessive activation, the model tends to rely more heavily on its internal knowledge, consequently producing unfaithful outputs.

Building on this observation, we propose **Param**etric Knowledge **Mut**ing through FFN Suppression (ParamMute), a novel framework designed to enhance the contextual faithfulness of LLMs. Specifically, ParamMute first identifies the FFN layers most associated with unfaithful generation and suppresses their activation to mitigate the undue influence of internal knowledge. A plug-and-play knowledge preference calibration module is then applied to the suppressed LLM to further encourage reliance on external evidence, ultimately yielding more trustworthy responses.

Additionally, to reliably evaluate LLM faithfulness, we introduce CoFaithfulQA, a comprehensive benchmark built from six open-domain QA datasets. It focuses on realistic scenarios where model responses may conflict with accurate retrieved evidence. Experimental results demonstrate that ParamMute consistently outperforms strong baselines on both CoFaithfulQA and the established ConFiQA benchmark [1]. It improves faithfulness by an average of 6.17% and 54.63% on the two benchmarks, respectively, while substantially reducing reliance on parametric knowledge. These results highlight the importance of explicitly accounting for internal knowledge as a key step toward building more faithful and trustworthy language models.

## 2 Preliminaries: Understanding the Role of FFN in Unfaithful Generation

In this work, we aim to investigate the influence of internal knowledge on unfaithful generation. To explore this, we focus on feed-forward networks, which interpretability studies have identified as primary repositories of parametric knowledge [21, 74]. This makes them ideal targets for analyzing the role of internal knowledge in unfaithful generation. This section begins by outlining the foundational concepts of knowledge representation and neuron activation in LLMs. We then conduct an empirical analysis using FFN activation patterns as a proxy for internal knowledge utilization, aiming to investigate their correlation with unfaithful model outputs.

### 2.1 Background: FFNs as Knowledge Carriers and Activation Analysis

**Feed-Forward Networks as Parametric Knowledge Stores.** Recent interpretability studies have shown that FFNs function similarly to key-value memory mechanisms, storing the majority of the parametric knowledge [20] through two parameter matrices $\boldsymbol{K}, \boldsymbol{V} \in \mathbb{R}^{d_m \times d}$, where $d_m$ and $d$ are the dimensions of the intermediate and input representations, respectively. For the $i$-th token in the input sequence, the FFN processes its representation $\boldsymbol{x}_i \in \mathbb{R}^d$ from the last layer through linear transformations. Formally, the computation in the $l$-th FFN can be expressed as a key-value memory mechanism:

$$\text{FFN}(\boldsymbol{x}_i^l) = (\sigma(\boldsymbol{K}^l \boldsymbol{x}_i^l))^\top \boldsymbol{V}^l, \tag{1}$$

where $\sigma$ is the activation function. Geva et al. [20] further show that the FFN output can be expressed as a weighted sum over a set of value vectors:

$$\text{FFN}(\boldsymbol{x}_i^l) = \sum_{j=1}^{d_m} \sigma(\boldsymbol{x}_i^l \cdot \boldsymbol{k}_j^l) \boldsymbol{v}_j^l = \sum_{j=1}^{d_m} a_{ij}^l \boldsymbol{v}_j^l, \tag{2}$$

where $\boldsymbol{k}_j^l$ and $\boldsymbol{v}_j^l$ denote the $j$-th row of $\boldsymbol{K}^l$ (the subkey) and the $j$-th column of $\boldsymbol{V}^l$ (the subvalue), respectively. The term $a_{ij}^l = \sigma(\boldsymbol{x}_i^l \cdot \boldsymbol{k}_j^l)$ represents the *activation coefficient* associated with the neuron $\boldsymbol{v}_j^l$. Following Mu et al. [47], we consider a neuron *activated* when $a_{ij}^l$ exceeds zero.

**Activation-based Metric.** Since each activated FFN neuron contributes independently to the final output [20, 21], we can quantify the overall activation level through an *activation ratio*. For a token representation $\boldsymbol{x}_i^l$ at layer $l$, the activation ratio $R^l(\boldsymbol{x}_i^l)$ at layer $l$ is defined as the fraction of neurons that are activated:

$$R^l(x_i^l) = \frac{1}{d_m} \sum_{j=1}^{d_m} \mathbb{I}[a_{ij}^l], \tag{3}$$

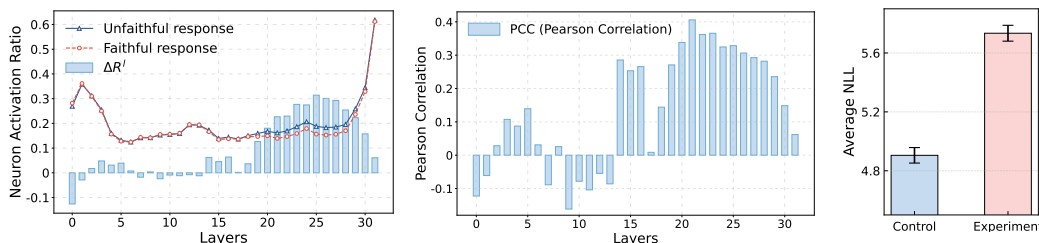

(a) Difference in Neuron Activation Ratio: Faithful vs. Unfaithful.
(b) Pearson Correlation: Neuron Activation Ratio vs. Unfaithful Label.
(c) UA-FFNs suppression increases NLL.

Figure 1: **Activation Pattern Differences and Causal Impact on Unfaithfulness.** (a) Activation ratio comparison between faithful and unfaithful generations. (b) Pearson correlation between unfaithfulness and FFN activation ratio, with UA-FFNs layers highlighted. (c) Suppressing UA-FFNs increases the Negative Log-Likelihood Loss (NLL) on unfaithful data, indicating a causal role.

where $\mathbb{I}[a_{ij}^l]$ is an indicator function that returns 1 if $a_{ij}^l > 0$, and 0 otherwise. Intuitively, a higher $R^l(x_i^l)$ indicates that more neurons in the FFN are actively participating in computing the output, reflecting a greater involvement of parametric knowledge stored in the FFN layer [15, 20, 74]. We compute the response-level activation ratio by averaging the activation ratios over all tokens in the response $\hat{r} = \{r_1, \ldots, r_T\}$:

$$R^l(\hat{r}) = \frac{1}{T} \sum_{i=1}^{T} R^l(r_i^l). \tag{4}$$

## 2.2 Pilot Study: Are Certain FFNs Implicated in Unfaithful Generation?

Building on the activation-based analysis framework introduced above, we now conduct an empirical investigation into a key hypothesis: *Do unfaithful responses correspond to disproportionately high activation in certain FFN layers?*

**Dataset for Activation Analysis.** To support this analysis, we use the proposed benchmark CoFaithfulQA, denoted as $\mathcal{D}$, which consists of model-generated responses annotated with binary faithfulness labels. These annotations enable direct comparison of activation patterns between faithful and unfaithful generations. Each instance $(q, c, y^*, \hat{r}, y_f) \in \mathcal{D}$ includes an input query $q$, a retrieved context $c$, a ground-truth answer $y^*$ derived from the evidence $c$, a model-generated response $\hat{r}$, and a binary label $y_f \in \{0, 1\}$, indicating whether $\hat{r}$ is faithful to the context $c$ (see Section 4 for construction and annotation details). For comparative analysis, we partition $\mathcal{D}$ into a faithful subset $\mathcal{D}^+$ and an unfaithful subset $\mathcal{D}^-$ based on the faithfulness label $y_f$. We then analyze the FFN activation patterns of the LLaMA3-8B-Instruct model across the two groups to investigate how activation behavior differs between faithful and unfaithful generations.

**Activation Differences Between Faithful and Unfaithful Responses.** To quantitatively examine the relationship between FFN activation and response faithfulness, we compute the layer-wise activation ratio $R^l(\hat{r})$, as defined in Eq. 4, for both the unfaithful subset $\mathcal{D}^-$ and the faithful subset $\mathcal{D}^+$. We then define their difference as the *activation gap*, given by:

$$\Delta R^l = \mathbb{E}_{\mathcal{D}^-}[R^l(\hat{r})] - \mathbb{E}_{\mathcal{D}^+}[R^l(\hat{r})] \tag{5}$$

As shown in Figure 1(a), while most FFN layers exhibit minimal differences, we observe consistently higher activation in $\mathcal{D}^-$ within a narrow range of layers, particularly in the middle-to-deep transformer blocks. This pattern suggests that unfaithful generations may be associated with distinct activation behaviors concentrated in these specific layers (robust across diverse settings; see Appendix A.4).

**Correlation and Causal Analysis of FFN Activation for Unfaithful Generation.** To statistically verify this association, we compute the Pearson Correlation Coefficient (PCC) between the activation ratio $R^l(\hat{r})$ and the unfaithfulness indicator $(1 - y_f)$ across the dataset. As shown in Figure 1(b), mid-to-deep FFN layers exhibit increasingly positive correlations (p-value < 0.05), confirming a significant positive correlation between activation in these layers and unfaithful generation. This evidence supports our hypothesis that a specific subset of mid-to-deep FFN layers—termed *Unfaithfulness-Associated FFNs (UA-FFNs)*—plays a central role in unfaithful generation. When these layers exhibit

excessive activation, the model increasingly relies on internal parametric knowledge (as further evidenced in Appendix A.15), overriding retrieved context and leading to unfaithful outputs.

To examine whether the observed correlation reflects a causal relationship, we perform a causal intervention [17] by suppressing the activation of selected FFN layers. Specifically, we compare the Negative Log-Likelihood (NLL) loss between an experimental group (with suppressed UA-FFNs) and a control group (using the vanilla model) on the unfaithful subset $\mathcal{D}^-$. The detailed intervention procedures are described in Appendix A.3. As shown in Figure 1(c), the experimental group exhibits consistently higher NLL than the control group, as expected—indicating that suppression of UA-FFNs makes unfaithful responses harder to generate. These results provide causal evidence that the activation strength of UA-FFNs directly influences the likelihood of unfaithful generation.

**Summary and Implications.** Our pilot study reveals that unfaithful generation in LLMs is associated with the over-reliance on internal parametric knowledge through UA-FFNs. Motivated by this, ParamMute (§3) applies selective suppression to UA-FFNs activations to limit parametric knowledge expression and improve contextual faithfulness.

# 3 Methodology

In this section, we present **Param**etric Knowledge **Mut**ing through FFN Suppression (ParamMute), a two-stage framework for improving the contextual faithfulness of LLMs. ParamMute first mitigates the influence of parametric knowledge by suppressing the activation of UA-FFNs (§3.1), and then incorporates an adaptation module to promote reliance on external knowledge (§3.2).

## 3.1 Reducing Internal Knowledge Reliance via Activation Suppression

Our pilot study in Section 2.2 reveals that unfaithful responses tend to involve a greater degree of internal parametric knowledge within a specific subset of FFNs (UA-FFNs). Motivated by this finding, we propose to suppress the activation of UA-FFNs, aiming to reduce the influence of internal knowledge and thereby enhance contextual faithfulness. Formally, for each layer $l$, we compute the average activation ratio $R^l(\hat{r})$ on both the unfaithful subset $\mathcal{D}^-$ and the faithful subset $\mathcal{D}^+$. We then use the previously defined activation gap $\Delta R^l$ (Eq. 5) to capture the difference in FFN activation between unfaithful and faithful outputs. Finally, we rank all layers $\mathbb{L}$ by their corresponding $\Delta R^l$ and select the top-$N$ layers with the highest activation gaps for subsequent suppression:

$$L_{\text{sup}} = \{l \in \mathbb{L} \mid l \text{ ranks in Top-}N \text{ of } \Delta R^l\}. \tag{6}$$

A suppression coefficient $\lambda \in [0, 1]$ is introduced to reduce the activation of UA-FFNs. Accordingly, the original FFN computation (Eq. 1) is reformulated as:

$$\text{FFN}^l(\boldsymbol{x}_i^l) = \left(\lambda \cdot \sigma(\boldsymbol{K}^l \boldsymbol{x}_i^l)\right)^\top \boldsymbol{V}^l, \quad \text{if } l \in L_{\text{sup}}. \tag{7}$$

Setting $\lambda = 1$ restores the model's original behavior. As $\lambda$ decreases, the contribution of the selected FFNs is progressively reduced. When $\lambda = 0$, the suppressed FFNs are fully deactivated and no longer influence the model's output. This soft suppression mechanism enables fine-grained control over the contribution of internal parametric knowledge (see Appendix A.14 for a detailed experiment and analysis).

## 3.2 Knowledge-Augmented Adaptation through Preference Optimization

After suppression, we further incorporate a plug-and-play adaptation module [24] to recalibrate the model's knowledge utilization preferences, enabling more effective usage of external evidence.

$$\mathcal{L} = \sum_D \alpha \cdot \mathcal{L}_{\text{KAT}} + \beta \cdot \mathcal{L}_{\text{KPO}}, \tag{8}$$

where $D$ denotes the set of all training instances, each comprising a query $q$, a retrieved context $c$, and a ground-truth answer $y^*$; and $\alpha$ and $\beta$ are hyperparameters that control the balance between the two objectives. The Knowledge-Augmented Training ($\mathcal{L}_{\text{KAT}}$) and Knowledge Preference Optimization ($\mathcal{L}_{\text{KPO}}$) objectives guide the suppressed model to both generate accurate answers and calibrate its knowledge usage preference towards external knowledge.

**Knowledge-Augmented Finetuning.** Following Lin et al. [40], we maximize the likelihood of generating the ground truth answer $y^*$ based on both query $q$ and external knowledge $c$:

$$\mathcal{L}_{\text{KAT}} = -\log P(y^* \mid q, c), \tag{9}$$

This objective trains the suppressed model to leverage both internal and external knowledge to answer the question accurately.

**Knowledge Preference Optimization.** To further refine the model's reliance on external versus internal knowledge, we apply a max-margin loss [11] to optimize the likelihood of generating ground truth answers that depend more on external knowledge:

$$\mathcal{L}_{\text{KPO}} = [\gamma - \log P(y^* \mid q, c) + \log P(y^* \mid q)]_+ , \tag{10}$$

where $\gamma$ is a margin parameter that controls the preference constraint, and the $[\cdot]+$ function ensures non-negativity. This objective further finetunes the suppressed model to shift its reliance towards external knowledge, improving the reliability and faithfulness of its responses.

## 4 CoFaithfulQA: A Consistency-Filtered Contextual Faithfulness QA Dataset

In this section, we introduce **Co**nsistency-filtered Contextual **Faithful**ness **QA** (CoFaithfulQA), a benchmark designed to evaluate the faithfulness of LLMs, and present the data collection pipeline along with the manual effort involved in constructing CoFaithfulQA.

**Characteristics of CoFaithfulQA.** Evaluating contextual faithfulness requires scenarios in which external evidence should override a model's incorrect internal knowledge. However, prior work has primarily relied on synthetic counterfactual contexts that contradict known correct answers [43, 45, 57, 65]. While effective for controlled testing, such synthetic settings often fail to reflect the naturally occurring inconsistencies between retrieved evidence and model responses that commonly arise in real-world applications.

| Dataset | #Full* | #Faith. | #Unfaith. |
|---|---|---|---|
| HotpotQA | 5,901 | 1,546 | 1,427 |
| NewsQA | 4,212 | 374 | 886 |
| NQ | 7,314 | 3,010 | 572 |
| SearchQA | 16,980 | 10,692 | 1,441 |
| SQuAD | 10,490 | 2,799 | 2,225 |
| TriviaQA | 7,785 | 5,887 | 767 |

Table 1: Statistics of the CoFaithfulQA dataset. **#Full\*** denotes the number of deduplicated examples from the original dataset. **#Faith.** and **#Unfaith.** indicate the number of samples labeled as faithful and unfaithful, corresponding to $\mathcal{D}^+$ and $\mathcal{D}^-$, respectively.

**Data Collection and Processing Pipeline.** Our CoFaithfulQA is constructed from six widely-used open-domain QA datasets: Natural Questions (NQ) [33], SQuAD [51], NewsQA [60], TriviaQA [31], SearchQA [13], and HotpotQA [69]. These datasets span a diverse range of domains, question types, and reasoning requirements, collectively forming a comprehensive evaluation testbed. Each QA triplet $(q, c, y^*) \in \mathcal{D}$ consists of the query $q$, relevant evidence $c$ and the ground truth $y^*$. Then the QA triplet is augmented as $(q, c, y^*, \hat{r}, y_f)$ with a model-generated response $\hat{r}$ and a faithfulness label $y_f$, based on which we form the subsets of CoFaithfulQA, $\mathcal{D}^-$ ($y_f = 0$) and $\mathcal{D}^+$ ($y_f = 1$).

To facilitate the evaluation of LLM faithfulness, CoFaithfulQA is constructed to reflect realistic scenarios where models are expected to rely on accurate external evidence rather than incorrect parametric knowledge. Specifically, we employ a two-stage pipeline: we first extract the model's dominant parametric knowledge through self-consistency filtering, and then identify conflicts between this belief and retrieved evidence using multi-model verification. This procedure ensures that the resulting dataset captures genuine failures of contextual faithfulness.

*Parametric Knowledge Elicitation.* We adopt a closed-book QA setup and apply a self-consistency mechanism [61, 44] to robustly capture the model's parametric knowledge. Specifically, we prompt the model $n$ times with the same query and designate the most frequently generated answer (i.e., the majority answer, denoted as $\hat{r}$) as its dominant belief. Queries for which the majority answer appears fewer than $n/2$ times are discarded to ensure consistency and reliability. Appendix A.7 provides evidence that higher self-consistency improves the quality of faithfulness assessment.

*Conflict Detection.* To identify whether the model's parametric knowledge contradicts the external evidence, we compare the dominant answer $\hat{r}$ with the ground-truth answer and the retrieved context. Two advanced pretrained LLMs—GPT-4o [49] and GLM-4-plus [22]—are used to assess whether a conflict exists. To mitigate model-specific bias, only instances where both models agree on the

presence of a conflict are retained. Based on this judgment, we assign a faithfulness label $y_f \in \{0, 1\}$, where $y_f = 0$ indicates that $\hat{r}$ conflicts with the context, and $y_f = 1$ otherwise. Appendix A.5 details the implementation procedure. Furthermore, we manually verify a subset of the detected conflicts to confirm their validity against human annotations (see Appendix A.6).

# 5 Experimental Methodology

This section describes datasets, evaluation metrics, baselines and implementation details.

**Datasets.** We evaluate the contextual faithful generation performance of different models on the subset $\mathcal{D}^-$ of CoFaithfulQA, as $\mathcal{D}^+$ samples are already contextually faithful and thus less informative for evaluation. To ensure a comprehensive evaluation, we also include two out-of-domain benchmarks: ConFiQA [1] and FaithEval [45]. ConFiQA assesses faithfulness in counterfactual scenarios across three subsets—Question Answering, Multi-hop Reasoning, and Multi-Conflicts—each containing 6,000 carefully constructed instances. Similarly, FaithEval tests a model's ability to prioritize the given text over its parametric knowledge.

**Evaluation.** Following Longpre et al. [43], we adopt a suite of metrics to evaluate the contextual faithfulness of model outputs. To ensure comparability, both generated responses and reference answers are normalized using the approach of Li et al. [39]. We report two primary metrics: context recall (ConR↑), which reflects the degree to which the model's responses align with the provided external context, and memory recall (MemR↓), which indicates reliance on the model's internal parametric knowledge. To further characterize the model's preference between these two sources, we also report the memorization ratio, defined as $\mathrm{MR} = \frac{\mathrm{MemR}}{\mathrm{MemR}+\mathrm{ConR}}$, which quantifies the model's relative tendency to favor memorized content over retrieved evidence.

**Baselines.** We evaluate ParamMute against a range of competitive baselines, categorized into four groups: (1) *Prompt-based approaches*, including the attributed prompt ($\mathrm{Attr_{prompt}}$) and the combined opinion-based and instruction-based prompt ($\mathrm{O\&I_{prompt}}$) from Zhou et al. [79]; (2) *Decoding-based methods*, where we select the representative COIECD [75], which incorporates entropy-based constraints to perform context-aware contrastive decoding; (3) *Fine-tuning methods*, consisting of standard Supervised Fine-Tuning (SFT) and Knowledge Aware Fine-Tuning (KAFT) [37]. KAFT enhances context faithfulness through counterfactual data augmentation; and (4) *Alignment-based methods*, including Context-DPO (C-DPO) [1], which applies the DPO framework [50] to encourage context-grounded responses while penalizing reliance on parametric memory, and DDR [39], which incorporates differentiable data rewards to train models to better use contextual knowledge.

**Implementation Details.** To ensure a fair comparison, we use LLaMA3-8B-Instruct as the backbone model for all methods throughout our experiments. Our configuration for ParamMute involves suppressing $N = 8$ UA-FFNs with a coefficient of 0.0 (Eq. 11). Notably, this specific set of layers is kept consistent across all datasets. The hyperparameters $\alpha$ and $\beta$, which balance $\mathcal{L}_{\mathrm{KAT}}$ and $\mathcal{L}_{\mathrm{KPO}}$ in Eq. 8, are both set to 0.5. Additional implementation details for our method and the baselines are provided in Appendix A.8 and A.9. Furthermore, we report a detailed failure case analysis in Appendix A.11 and results on different backbone models in Appendix A.16.

# 6 Experiment Results

In this section, we first present the overall performance of ParamMute (§6.1), followed by a comprehensive ablation study (§6.2), a detailed parameter sensitivity analysis (§6.3), and an investigation into how ParamMute calibrates the knowledge utilization of LLMs (§6.2).

## 6.1 Main Results

This experiment evaluates ParamMute on CoFaithfulQA to assess its overall performance. Additionally, we test ParamMute on the ConFiQA dataset, which represents an out-of-domain setting.

As shown in Table 2, ParamMute significantly outperforms baseline models on CoFaithfulQA, demonstrating its effectiveness in generating more accurate and contextually faithful responses. Compared to the vanilla RAG model, ParamMute achieves an average improvement of 5% in ConR and reduces MemR by 4%, effectively mitigating the model's reliance on parametric knowledge

| Models | HotPotQA | | | NQ | | | NewsQA | | |
|---|---|---|---|---|---|---|---|---|---|
| | ConR ↑ | MemR ↓ | MR ↓ | ConR ↑ | MemR ↓ | MR ↓ | ConR ↑ | MemR ↓ | MR ↓ |
| Vanilla-RAG [52] | 60.34 | 13.88 | 18.70 | 53.09 | 14.41 | 21.35 | 60.27 | 8.24 | 12.03 |
| Attr$_{prompt}$ [79] | 58.93 | 13.95 | 19.13 | 55.36 | 11.07 | 16.67 | 58.80 | 7.56 | 11.39 |
| O&I$_{prompt}$ [79] | 47.79 | 10.72 | 18.32 | 49.25 | **8.23** | 14.32 | 52.03 | 5.30 | 9.25 |
| COIECD [75] | 62.51 | 12.19 | 16.32 | 56.21 | 12.28 | 17.93 | 51.81 | 6.21 | 10.70 |
| SFT [63] | 70.92 | 6.24 | 8.08 | 59.76 | 10.29 | 14.69 | 61.96 | 5.08 | 7.58 |
| KAFT [37] | 69.52 | 6.87 | 8.99 | 60.89 | 9.23 | 13.16 | 65.09 | **4.74** | **6.79** |
| C-DPO [1] | 67.20 | 7.64 | 10.21 | **62.24** | 9.79 | 13.6 | 61.4 | **4.74** | 7.17 |
| DDR [39] | 68.66 | 7.15 | 9.43 | **63.29** | 10.33 | 14.03 | 64.74 | 5.03 | 7.21 |
| ParamMute | **71.06** | **6.17** | **7.99** | 60.68 | 9.08 | **13.02** | 65.24 | 4.85 | 6.92 |

| Models | SearchQA | | | SQuAD | | | TriviaQA | | |
|---|---|---|---|---|---|---|---|---|---|
| | ConR ↑ | MemR ↓ | MR ↓ | ConR ↑ | MemR ↓ | MR ↓ | ConR ↑ | MemR ↓ | MR ↓ |
| Vanilla-RAG [52] | 66.76 | 10.55 | 13.64 | 77.93 | 6.79 | 8.01 | 61.80 | 11.47 | 15.66 |
| Attr$_{prompt}$ [79] | 62.53 | 10.55 | 14.43 | 77.35 | 6.38 | 7.62 | 59.97 | 10.43 | 14.81 |
| O&I$_{prompt}$ [79] | 52.26 | 9.23 | 15.01 | 76.81 | 6.11 | 7.37 | 55.41 | 8.08 | 12.73 |
| COIECD [75] | 69.74 | 11.66 | 14.32 | 73.12 | 7.64 | 9.46 | **63.62** | 11.99 | 15.86 |
| SFT [63] | 75.29 | 6.87 | 8.36 | 79.19 | 4.22 | 5.06 | 59.6 | 8.34 | 12.38 |
| KAFT [37] | 77.38 | 7.43 | 8.76 | 80.04 | 4.18 | 4.96 | 62.32 | 8.74 | 12.29 |
| C-DPO [1] | 64.12 | **5.62** | 8.06 | 80.08 | 5.26 | 6.16 | 58.67 | 8.74 | 12.96 |
| DDR [39] | 78.07 | 6.48 | 7.66 | **81.36** | 4.75 | 5.52 | 60.71 | 7.73 | 11.29 |
| ParamMute | **78.76** | 6.04 | **7.12** | 80.58 | **4.04** | **4.78** | 60.89 | **6.91** | **10.19** |

Table 2: Performance on the CoFaithfulQA dataset. The highest scores are highlighted in **bold**, while the second-highest scores are underlined.

| Models | Question Answering | | | Multi-hop Reasoning | | | Multi-Conflicts | | |
|---|---|---|---|---|---|---|---|---|---|
| | ConR ↑ | MemR ↓ | MR ↓ | ConR ↑ | MemR ↓ | MR ↓ | ConR ↑ | MemR ↓ | MR ↓ |
| Vanilla-RAG [52] | 26.24 | 38.51 | 59.47 | 14.87 | 24.98 | 62.69 | 4.49 | 13.53 | 75.09 |
| Attr$_{prompt}$ [79] | 47.33 | 25.78 | 35.26 | 17.69 | 22.42 | 55.90 | 6.60 | 14.67 | 68.97 |
| O&I$_{prompt}$ [79] | 66.22 | 13.69 | 17.13 | 16.78 | 17.18 | 50.59 | 11.64 | 12.60 | 51.97 |
| COIECD [75] | 71.69 | 15.33 | 17.62 | 53.36 | 17.13 | 24.31 | 57.11 | 9.60 | 14.39 |
| SFT [63] | 78.02 | 5.02 | 6.05 | 61.40 | 13.47 | 17.99 | 61.98 | 9.54 | 13.34 |
| KAFT [37] | **82.04** | 5.58 | 6.36 | **63.71** | 13.64 | 17.63 | **67.31** | 9.98 | 12.91 |
| C-DPO [1] | 81.82 | 6.20 | 7.04 | 58.89 | 14.00 | 19.21 | 58.24 | **8.71** | 13.01 |
| DDR [39] | 80.71 | 6.07 | 6.99 | 60.64 | 15.60 | 20.46 | 61.07 | 8.93 | 12.76 |
| ParamMute | 81.20 | **3.69** | **4.35** | 63.09 | **12.82** | **16.89** | 65.20 | 9.29 | **12.47** |

Table 3: Performance of different models on the testing sets of ConFiQA.

and encouraging better utilization of external context. The evaluation results also indicate that prompt-based methods and decoding-based approaches such as Attr$_{prompt}$, O&I$_{prompt}$, and COIECD decrease the MemR score, showing their effectiveness in reducing the model's reliance on parametric knowledge. However, they also lead to a decline in answer correctness, as reflected by lower ConR score compared to the Vanilla RAG model. In contrast, fine-tuning-based approaches, such as SFT, KAFT, DPO, and DDR, enhance contextual faithfulness by adjusting the parameters of LLMs, highlighting the crucial role these parameters play in the emergence of knowledge conflicts within the models. ParamMute usually shows better performance than these fine-tuning based methods, which thrives on its "suppression-and-adaptation" mechanism.

To further evaluate the generalization capability of ParamMute, we tested it on the ConFiQA and FaithEval datasets. As shown by the results (Table 3 for ConFiQA and Appendix A.13 for FaithEval), ParamMute outperforms both prompt-based and fine-tuning methods in enhancing contextual faithfulness and reducing reliance on parametric knowledge. These improvements highlight the effectiveness of ParamMute in encouraging LLMs to prioritize contextual evidence over internal memorization, demonstrating its strong generalization ability.

## 6.2 Understanding ParamMute via Ablation and Component Analysis

We conduct ablation studies to analyze the effectiveness of ParamMute's suppression strategy and to evaluate the contributions of its key components. Specifically, we compare suppression across

different model sublayers, examine alternative FFN selection strategies, and assess the individual impact of the suppression and adaptation modules.

**Are FFNs the Primary Drivers of Unfaithful Generation?** To assess the contribution of different transformer components to unfaithful generation, we evaluate different suppression strategies. In addition to suppressing the UA-FFNs sublayers identified by ParamMute, we evaluate three alternatives: suppressing multi-head attention sub-layers (MHA), suppressing knowledge-related parameters (Parameter) [34], and suppressing entire transformer layers (Layer). All strategies share the same implementation setup, except for the specific component being suppressed. Technical details are provided in Appendix A.10. As shown in Table 4 (rows 3–6), ParamMute yields the most significant improvements in contextual faithfulness. This suggests that FFN sublayers play a more central role in parametric knowledge recall than other components, consistent with prior findings that position FFNs as key repositories of internal memory [10, 21].

**Can Other FFNs Match the Effect of Those Selected by ParamMute?** To assess whether alternative FFN selections can achieve similar improvements in contextual faithfulness, we compare ParamMute with several variants that suppress different subsets of FFNs. Specifically, we experiment with suppressing FFNs in bottom layers, mid layers, and randomly selected layers, as detailed in Table 4 (rows 8–11). Our results show that suppressing bottom-layer FFNs leads to a substantial drop in ConR, indicating poor contextual grounding. Mid-layer and randomly selected FFNs suppressing methods yield moderately better performance, but still underperform ParamMute. These findings highlight the crucial role of the FFNs identified by ParamMute, underscoring their effectiveness in mitigating parametric knowledge reliance and improving contextual faithfulness.

| Method | ConR ↑ | MemR ↓ | MR ↓ |
|---|---|---|---|
| *Suppressed Component Selection* | | | |
| **FFN** | **69.54** | 6.18 | **8.34** |
| MHA | 68.35 | 6.81 | 9.23 |
| Layer | 62.52 | **6.03** | 8.92 |
| Parameter | 68.71 | 6.67 | 8.85 |
| *Suppressed Layer Selection* | | | |
| **UA–FFNs** | **69.54** | **6.18** | **8.34** |
| Bottom | 62.29 | 7.15 | 10.38 |
| Middle | 67.11 | 6.85 | 9.43 |
| Random | 67.65 | 7.20 | 9.84 |
| *Faithful Enhancement Strategies* | | | |
| **ParamMute** | **69.54** | **6.18** | **8.34** |
| w/o Suppression | 69.47 | 7.01 | 9.34 |
| w/o Adaption | 68.57 | 6.32 | 8.58 |

Table 4: Comparison of suppression strategies in ParamMute, covering component-level and layer-level variants, along with ablation studies on suppression and adaptation components.

**Contributions of Different Components of ParamMute.** As shown in Table 4 (rows 13-15), we compare ParamMute with two ablated variants: ParamMute w/o Suppression and ParamMute w/o Adaptation, in order to examine the contributions of each component. Removing the suppression module results in an increase of approximately 0.8% in MemR, suggesting that suppressing activation is effective in reducing reliance on parametric knowledge. In contrast, removing the Adaptation module leads to a 1% drop in ConR, highlighting its role in promoting better use of external context. These findings confirm the effectiveness of ParamMute in reducing the dependence of LLMs on internal memory for faithful generation.

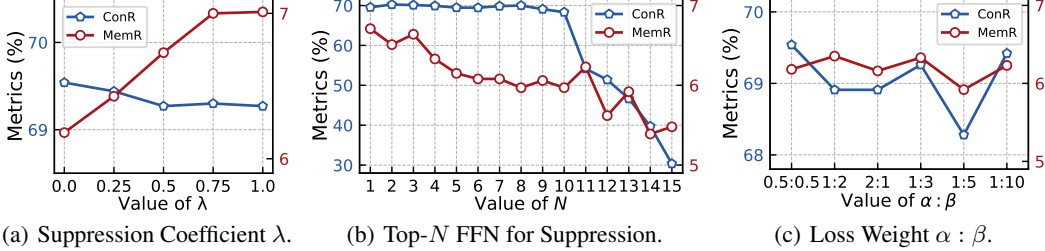

(a) Suppression Coefficient $\lambda$.  (b) Top-$N$ FFN for Suppression.  (c) Loss Weight $\alpha : \beta$.

Figure 2: Variation in ConR and MemR under different hyperparameter settings. Each point reflects the average metric across all subsets within CoFaithfulQA. Higher ConR and lower MemR indicate better contextual faithfulness with reduced parametric reliance.

## 6.3 Impact of Key Hyperparameters in ParamMute

To investigate the impact of key hyperparameters in ParamMute, we conduct a sensitivity analysis. The experimental setup remains identical to our main implementation, varying only the specific

hyperparameter under investigation. Specifically, we investigate three factors: (1) the suppression coefficient $\lambda$, which controls the strength of activation suppression applied to selected FFNs; (2) the number of top-$N$ FFNs selected for suppression; and (3) the weighting coefficients $\alpha$ and $\beta$ used to balance the $\mathcal{L}_{\text{KAT}}$ and $\mathcal{L}_{\text{KPO}}$ during training. The results are presented in Figure 2.

**Suppression Coefficient $\lambda$.** We vary $\lambda \in [0.0, 1.0]$ to analyze the impact of suppression strength applied to UA-FFNs activations, where $\lambda = 0.0$ denotes full suppression and $\lambda = 1.0$ corresponds to the original model without intervention. The same value for $\lambda$ is used consistently during both training and inference. As shown in Figure 2(a), decreasing $\lambda$ consistently reduces MemR and improves ConR, indicating that smaller $\lambda$ values lead to better contextual faithfulness and reduced reliance on internal memory. At $\lambda = 0.0$, the model achieves the best overall performance. Given its strong effect in promoting contextual faithfulness, we adopt $\lambda = 0.0$ as the default setting in all experiments unless otherwise specified.

**The Number of Suppressed FFNs ($N$).** We investigate how the number of top-activated FFNs selected for suppression affects the model's behavior. Specifically, we vary $N$ from 1 to 15, covering nearly half of all FFN layers. As shown in Figure 2(b), increasing $N$ expands the suppression scope and consistently reduces MemR. However, when $N$ reaches 10, we observe a sharp drop in ConR, suggesting that excessive suppression may interfere with functions beyond knowledge storage. This indicates that not all FFN layers are suppressible without adverse effects, and overly broad suppression can impair the model's ability to utilize external context.

**Loss Balancing Coefficients $\alpha$ and $\beta$.** During joint training, we use $\alpha$ and $\beta$ to weight Knowledge-Augmented Training ($\mathcal{L}_{\text{KAT}}$) and Knowledge Preference Optimization ($\mathcal{L}_{\text{KPO}}$), respectively. We empirically test different ratios of $\alpha : \beta$ and find that varying this ratio has limited impact on overall performance. Nonetheless, moderate weighting (e.g., $\alpha = 0.5$, $\beta = 0.5$) achieves a good balance between suppressing parametric interference and maintaining task accuracy (see Figure 2(c)).

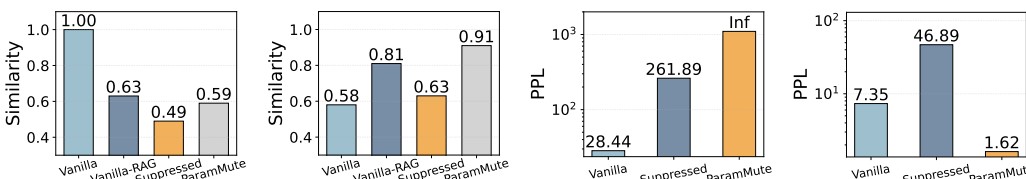

(a) Response Similarity with Parametric Answer. (b) Response Similarity with Contextual Answer. (c) Perplexity w/o Context. (d) Perplexity w/ Context.

Figure 3: Evaluation of knowledge utilization of different models. We assess the response similarity with parametric answer and contextual answer (Figure 3(a) and Figure 3(b)), and compute the PPL score when reproducing the ground truth answer (Figure 3(c) and Figure 3(d)). The Suppressed model refers to ParamMute w/o Adaption, which only incorporates the knowledge suppression.

### 6.4 Effectiveness of ParamMute in Calibrating Knowledge Usage Behavior

To assess whether ParamMute improves contextual faithfulness by guiding LLMs to favor retrieved evidence over incorrect internal knowledge, we conduct a comparative analysis on $\mathcal{D}^-$, the unfaithful subset of CoFaithfulQA. We compare the performance of three models: the vanilla LLM, the Suppressed model (ParamMute w/o Adaptation), and our ParamMute.

We first evaluate the model's knowledge usage preference by computing the semantic similarity between its outputs and two reference answers: (1) the parametric answer $\hat{r}$, representing the model's internal belief obtained in a closed-book setting (Section 4), and (2) the contextual answer $y^*$ derived from retrieved evidence. As shown in Figure 3(a), the Suppressed model achieves the lowest similarity to parametric answers, indicating that activation suppression effectively weakens reliance on internal knowledge. Meanwhile, Figure 3(b) shows that ParamMute achieves the highest similarity with contextual answers, indicating the effectiveness of ParamMute in enhancing the context knowledge usage ability of LLMs by using a plug-and-play knowledge adaptation module.

To further assess knowledge calibration, we measure the perplexity (PPL) of each model when reproducing the ground-truth answer, both with and without contextual input. A lower PPL indicates greater confidence in generating the correct response. Figure 3(c) shows that when no context is provided, the Suppressed model exhibits a higher PPL, confirming its effectiveness in reducing the

dependence on parametric memory. Alternatively, ParamMute displays extremely high PPL in the absence of context but significantly lower PPL when context is available (Figure 3(d)), confirming that the model has shifted to reliance primarily on retrieved evidence instead of the parametric knowledge.

# 7 Related work

Despite considerable advancements of Retrieval-Augmented Generation (RAG) models [52, 56, 70], unfaithful generation [25]—where models produce content that is not supported by, or even contradicts, the retrieved external evidence—remains a critical and persistent challenge. Even when supplied with accurate and relevant external knowledge, RAG models frequently prioritize their internal parametric knowledge over retrieved information, leading to unfaithful outputs and diminishing the reliability of such systems [3, 6, 7, 73, 65]. Thus, the demand for contextually faithful LLMs has significantly increased, particularly within RAG applications [4, 36, 5].

Numerous studies have systematically investigated this phenomenon from both evaluation and analytical perspectives. For instance, certain research constructs synthetic scenarios by manually replacing entities in retrieved passages, highlighting the propensity of LLMs to generate responses aligned with their internal knowledge rather than provided external evidence [29, 43, 45]. Other studies demonstrate that LLMs often opt for contextually plausible but internally memorized information when faced with conflicting sources, underscoring the difficulty of overcoming ingrained parametric knowledge biases [32, 65]. Additionally, Jin et al. [30] identifies separate context and memory attention heads, which respectively attend to external and internal sources of information, offering a more granular view into the mechanisms that underlie unfaithful generation. Complementarily, Sun et al. [58] suggest that certain FFNs within LLMs act as knowledge injectors, amplifying the influence of internal memory within the residual stream and thereby contributing to unfaithful generation.

Efforts to improve contextual faithfulness primarily focus on enhancing external knowledge integration through various strategies. One direction focuses on prompt design to guide models toward context-grounded responses [62, 79]. Another approach encompasses fine-tuning LLMs on knowledge-augmented datasets, reinforcing the model's preference for retrieved information over internal memory [16, 37, 46, 48]. Alignment techniques have also been explored, aiming to encourage external grounding while suppressing dependence on internal parametric knowledge [1, 39]. Moreover, contrastive decoding methods have been proposed, explicitly differentiating between faithful and hallucinated responses to promote alignment with external evidence during generation [2, 55, 29]. Beyond external interventions, prior work [30, 58] has also highlighted the role of internal components such as FFNs in shaping model behavior. Building on this, our work analyzes FFN activation patterns to identify over-active layers strongly correlated with unfaithful outputs. We propose a suppression-based strategy to reduce their influence and enhance contextual grounding.

# 8 Conclusion

In this paper, we introduce ParamMute, a novel framework designed to enhance the contextual faithfulness of LLMs. Our approach addresses the persistent challenge of LLMs favoring internal parametric knowledge over retrieved evidence. ParamMute first mitigates this over-reliance by strategically suppressing the activation of specific FFNs that exhibit a strong correlation with unfaithful generation. To further promote adherence to external information, ParamMute incorporates a plug-and-play adaptation module that reinforces the model's grounding in the retrieved content. Additionally, we introduce CoFaithfulQA, a comprehensive benchmark constructed from six diverse QA datasets, enabling controlled evaluation of faithfulness under conflicting knowledge settings. Extensive experiments on CoFaithfulQA and ConFiQA demonstrate that ParamMute significantly enhances generation faithfulness while substantially mitigating dependence on internal knowledge.

# 9 Acknowledgment

This work is partly supported by the National Natural Science Foundation of China (No. 62206042 and No. 62461146205), CCF-zhipu Large Model Innovation Fund (No. 202403), and the Fundamental Research Funds for the Central Universities (No. N25ZLL045). This work is also supported by the AI9Stars community.

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

# A  Appendix

## A.1  License

We are committed to ethical research practices and ensuring the reproducibility of our work. To this end, we present the licensing information for all datasets utilized in our experiments. These include Natural Questions (CC BY-SA 3.0), NewsQA (MIT License), SearchQA (Apache License 2.0), TriviaQA (Apache License 2.0), HotpotQA (CC BY-SA 4.0), and SQuAD (CC BY-SA 4.0). Our use of these datasets is in full compliance with their terms, as all aforementioned licenses expressly permit their application in academic research.

## A.2  Ethics Statement

Our data construction process involves prompting LLMs to elicit their internal parametric knowledge in order to investigate the underlying causes of hallucinations in generated outputs. While this approach enables targeted analysis of model behavior, it may lead to the generation of inaccurate or hallucinated contents. To ensure the responsible usage, we strictly limit the distribution of the resulting dataset to academic research purposes. The dataset does not contain any personally identifiable information or offensive material, and all contents are curated in accordance with ethical guidelines for responsible AI research and data sharing.

Additionally, we conducted human evaluations to assess the reliability of the LLMs in identifying knowledge conflicts. Evaluation data was carefully distributed to human evaluators solely for research purposes, ensuring it adheres to ethical standards and contains no content that violates these standards. We also recognize that the capacity to suppress a model's knowledge raises important ethical concerns about its potential for misuse, such as obscuring facts or entrenching bias. Our work explores this mechanism purely as a means to improve contextual faithfulness and mitigate hallucinations. We stress that the transition of such capabilities from academic research to real-world deployment would require rigorous oversight and transparent governance to ensure responsible use.

## A.3  Causal Intervention on UA-FFNs Activation

To establish the causal role of UA-FFNs activation in unfaithful generation, we perform intervention experiments by manipulating the activation strength of the Unfaithfulness-Associated FFNs (UA-FFNs). These FFNs are identified in Section 2 as exhibiting strong correlations with unfaithful outputs. Our goal is to examine whether suppressing or enhancing their activation causally affects the faithfulness of the model's generation.

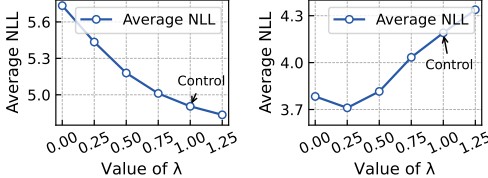

(a) Unfaithful subset $\mathcal{D}^-$.  (b) Faithful subset $\mathcal{D}^+$.

Figure 4: Average NLL loss under different FFN activation scales ($\lambda$) for an unfaithful subset $\mathcal{D}^-$ and a faithful subset $\mathcal{D}^+$.

**Intervention Setup.** We conduct our intervention experiments on the CoFaithfulQA using the LLaMA3-8B-Instruct model. Each instance $(q, c, y^*, \hat{r}, y_f) \in \mathcal{D}$ is labeled as faithful ($y_f$=1) or unfaithful ($y_f$=0), allowing us to partition the data into $\mathcal{D}^+$ and $\mathcal{D}^-$ for subsequent analysis. To modulate the influence of parametric knowledge, we apply a scaling factor $\lambda$ to the output of the selected UA-FFNs layers:

$$\text{UA-FFN}^l(\boldsymbol{x}_i^l) = \left(\lambda \cdot \sigma(\boldsymbol{K}^l \boldsymbol{x}_i^l)\right)^\top \boldsymbol{V}^l. \tag{11}$$

Here, $\lambda$ controls the activation of each UA-FFNs layer: when $\lambda < 1$, the contribution of parametric knowledge is suppressed; when $\lambda > 1$, it is amplified. To evaluate the model's sensitivity to such interventions, we vary $\lambda$ across $\{0.0, 0.25, 0.5, 0.75, 1.0, 1.25\}$. The unmodified model with $\lambda = 1.0$ serves as the control group, while all other settings constitute the experimental group.

**Evaluation Protocol.** We evaluate the effect of suppression on model behavior by computing the average negative log-likelihood (NLL) loss over two disjoint subsets of the dataset: the faithful subset $\mathcal{D}^+$ and the unfaithful subset $\mathcal{D}^-$. For each setting of the suppression coefficient $\lambda \in [0.0, 1.0]$, we measure the model's NLL loss separately on both subsets. The suppression is applied to UA-FFNs with varying $\lambda$, where $\lambda = 0.0$ denotes full suppression and $\lambda = 1.0$ corresponds to no suppression.

**Results.** Figure 4 summarizes the model behavior across a range of suppression coefficients $\lambda$. The endpoints, $\lambda = 0.0$ (full suppression) and $\lambda = 1.0$ (no suppression), correspond to the intervention and control settings introduced earlier in Figure 1(c) (Section 2.2).

When evaluated on the unfaithful subset $\mathcal{D}^-$, the NLL increases monotonically as $\lambda$ decreases, with the highest value observed under full suppression ($\lambda = 0.0$). This trend indicates that suppressing UA-FFNs activation effectively disrupts the model's ability to generate unfaithful responses, suggesting that these FFNs play a functional role in facilitating hallucinated content. Meanwhile, on the faithful subset $\mathcal{D}^+$, the NLL also decreases as $\lambda$ decreases. This trend suggests that suppressing UA-FFNs activation not only avoids harming faithful generation, but may even improve it. A possible explanation is that reducing reliance on parametric knowledge encourages the model to more effectively utilize the retrieved context, resulting in more faithful and confident responses. To further validate this trend, we increase the suppression coefficient to $\lambda = 1.25$, thereby amplifying the activation of UA-FFNs. As shown in Figure 4, this leads to a decrease in NLL on the unfaithful subset $\mathcal{D}^-$ and a moderate increase on the faithful subset $\mathcal{D}^+$. These findings further confirm that enhanced activation of UA-FFNs facilitates unfaithful generation.

**Results.** Figure 4 summarizes the model behavior across a range of $\lambda$ values. The endpoints–$\lambda = 0.0$ (full suppression) and $\lambda = 1.0$ (no suppression)–correspond to the intervention and control settings shown earlier in Figure 1(c) (Section 2.2). To better understand the effect of suppression strength, we examine model performance on the two subsets separately. For the unfaithful subset $\mathcal{D}^-$, we observe a consistent increase in NLL loss as $\lambda$ decreases, with a peak at $\lambda = 0.0$. This monotonic trend confirms that suppressing UA-FFNs activation disrupts the model's ability to produce hallucinated content, implying that these FFNs play a functional role in facilitating unfaithful generation. In contrast, the loss on the faithful subset $\mathcal{D}^+$ shows only a mild increase as $\lambda$ decreases, indicating that UA-FFNs contributes little to the generation when the model relies on retrieved context.

**Conclusion.** These results provide strong causal evidence that the over-activation of UA-FFNs drives unfaithful generation by injecting parametric knowledge into the output. By suppressing these layers, the model becomes less confident in producing hallucinated content, as reflected in the increased loss on $\mathcal{D}^-$. This confirms that internal memory representations in LLMs–particularly within specific FFNs–are not merely correlated with unfaithful generation, but actively responsible for their emergence.

| Dataset | Full Size* | Consistency | Faithful Subset ($\mathcal{D}^+$) | Unfaithful Subset ($\mathcal{D}^-$) |
|---|---|---|---|---|
| HotpotQA | 5,901 | 2,973 (50%) | 1,546 (26%) | 1,427 (24%) |
| NewsQA | 4,212 | 1,260 (30%) | 374 (9%) | 886 (21%) |
| NQ | 7,314 | 4,419 (60%) | 3,010 (41%) | 1,409 (19%) |
| SearchQA | 16,980 | 12,133 (71%) | 10,692 (63%) | 1,441 (8%) |
| SQuAD | 10,490 | 5,024 (48%) | 2,799 (27%) | 2,225 (21%) |
| TriviaQA | 7,785 | 6,654 (85%) | 5,887 (75%) | 767 (10%) |

Table 5: Number of instances at each stage in the CoFaithfulQA construction pipeline.

## A.4 Robustness of UA-FFN Patterns Across Diverse Settings

To test the generalizability of our finding—that excessive activation in a specific subset of mid-to-deep FFN layers causes the model to rely more on internal knowledge and produce unfaithful outputs—we extended our Pilot Study (§ 2.2) to different datasets, larger models, and models from different families. Specifically, to verify our finding's independence from the dataset, we evaluated LLaMA-3-8B-Instruct on HotpotQA and SQuAD. The results, shown in Figure 5(a) and Figure 5(b), combined with our initial findings on CoFaithfulQA (Figure 1(a)), confirm that our observation is dataset-agnostic. Furthermore, to assess generalizability across model scale and architecture, we experimented with Qwen-2.5-32B-Instruct and LLaMA-3-70B-Instruct. The outcomes (Figure A.3 and Figure A.3) revealed the same activation patterns identified in our pilot study.

The results consistently demonstrate that our key finding is robust: excessive activation in a specific subset of mid-to-deep FFN layers (typically 60%–85% of the model depth) is strongly associated with unfaithful outputs, regardless of dataset, model family, or size. This not only highlights the generality of our observation but also provides valuable insights into the mechanisms underlying

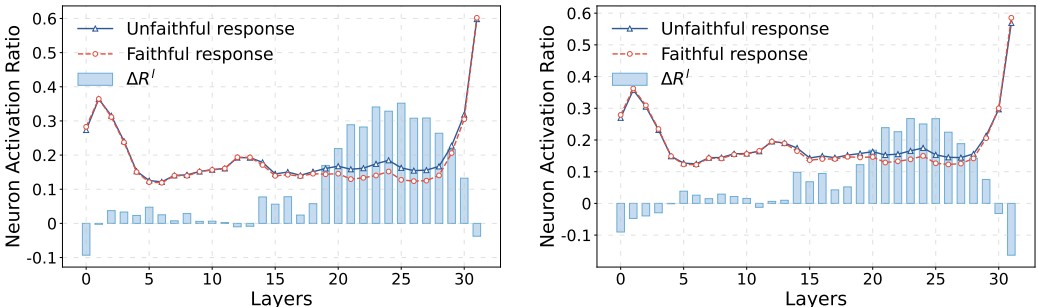

(a) Difference in Neuron Activation Ratio on Hot- (b) Difference in Neuron Activation Ratio on TriviaQA.
potQA.

Figure 5: Comparison of neuron activation ratios between faithful and unfaithful generations on HotpotQA and SQuAD, evaluated with LLaMA-3-8B-Instruct.

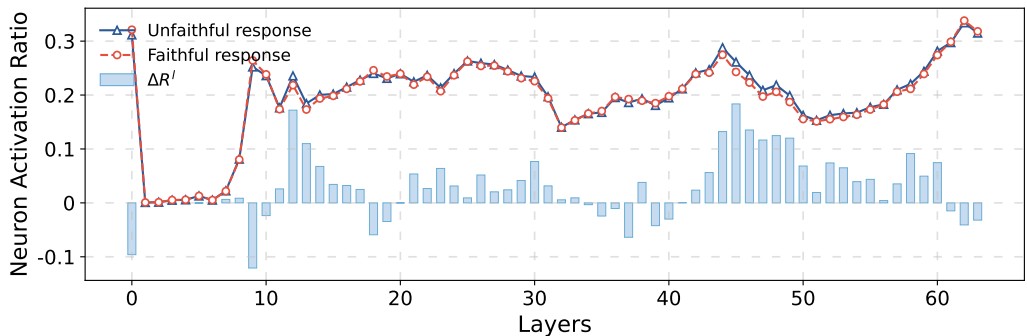

Figure 6: Comparison of neuron activation ratios between faithful and unfaithful generations on Qwen-2.5-32B-Instruct.

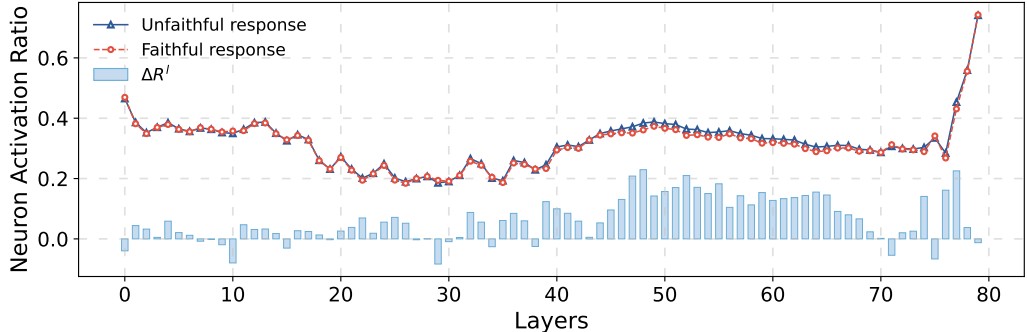

Figure 7: Comparison of neuron activation ratios between faithful and unfaithful generations on LLaMA-3-70B-Instruct.

unfaithful generation and the role of parametric knowledge in LLMs. We hope these findings will inform future research.

## A.5 Details of CoFaithfulQA Construction

In this section, we detail the two main steps in constructing CoFaithfulQA.

**Parametric Knowledge Elicitation.** To elicit the LLM's parametric knowledge, we prompt the model in a closed-book setting i.e., without providing any external context. To improve the reliability of the elicited responses, we adopt a consistency-based filtering strategy [68, 72]. For each query

$q$, the model is prompted $n = 5$ times, yielding a set of responses $\{r_1, r_2, \ldots, r_5\}$. We identify the majority response $\hat{r}$ as the one that appears most frequently. A query $q_i$ is retained if and only if the frequency of $\hat{r}$ is at least 3 (i.e., appears in $\geq 3$ out of 5 responses), thereby filtering out inconsistent generations and ensuring the reliability of the extracted parametric knowledge.

The following prompt template is used to elicit responses from the model:

---

**Prompt for Eliciting Parametric Knowledge**

Answer the question {*brevity_instruction*} and provide supporting evidence.
Question: {*question*}

---

The "*brevity_instruction*" is incorporated to encourage the LLM to produce more concise responses, following the guidance strategy proposed by Kortukov et al. [32].

**Conflict Detection.** Next, we categorize each instance obtained from the previous step into one of two groups–conflicting or non-conflicting–based on whether the model's parametric knowledge aligns with the retrieved context. To assess the presence of conflict, we employ LLMs to compare the parametric answer and the contextual evidence. To mitigate model-specific bias, we adopt a dual-model agreement strategy: a conflict label is only assigned when both GPT-4o [49] and GLM-4-plus [22] agree on its presence. For both models, we use the following prompt:

---

**Prompt for Identifying Conflict Knowledge**

You are tasked with evaluating the correctness of a model-generated answer based on the given information.
Context: {*context*}
Question: {*question*}
Contextual Answer: {*contextual_answer*}
Model-Generated Answer: {*Model-Generated_answer*}
[*Detailed task description...*]
Output Format:
Evaluate result: (Correct / Partially Correct / Incorrect)

---

Based on this process, we assign each instance an additional binary label $y_f$ indicating faithfulness: $y_f = 0$ (unfaithful) if the parametric knowledge conflicts with the context, and $y_f = 1$ (faithful) otherwise. The unfaithful subset $\mathcal{D}^-$ is used for downstream evaluation experiments, while the faithful subset $\mathcal{D}^+$ is used for activation analysis.

## A.6 Assessing the Reliability of LLMs in Knowledge Conflict Identification

In this subsection, we conduct a human evaluation to assess the reliability of GPT-4o and GLM-4-plus in identifying knowledge conflicts. This evaluation aims to verify whether LLMs can serve as trustworthy tools for automatically detecting conflicts between different knowledge sources, a critical step in our data construction pipeline.

To ensure broad coverage, we randomly sample 150 instances from each of the six subsets of CoFaithfulQA, resulting in a total of 900 examples that span diverse query types and conflict scenarios. Among them, 100 instances are randomly selected and independently annotated by multiple annotators to compute inter-annotator agreement (IAA). The annotations are conducted by six senior researchers (each holding at least a bachelor's degree) with backgrounds in computational linguistics and LLM behavior analysis, ensuring high-quality and consistent evaluations.

| Subset | Agreement (%) |
|---|---|
| HotpotQA | 89.4 |
| NewsQA | 91.3 |
| NQ | 89.2 |
| SearchQA | 94.6 |
| SQuAD | 87.5 |
| TriviaQA | 90.3 |
| **Average** | **90.4** |

Table 6: Agreement between human annotators and LLMs across different subsets of our CoFaithfulQA benchmark.

For each instance, annotators are provided with the question, the contextual answer, the model-generated response, and the corresponding supporting evidence. Unlike binary classification approaches (e.g., NLI-based models), we adopt a more fine-grained evaluation protocol. Annotators are asked to classify each response into one of three categories: *No Conflict*, *Somewhat Conflict*, or *High Conflict*. The detailed annotation instructions are as follows:

To ensure annotators fully understand the task, we first instruct them using a set of five gold-standard examples. Additionally, annotators had access to clarification support throughout the annotation process. We observe strong annotation consistency, with a Cohen's $\kappa$ of 0.766 between human annotators, indicating substantial inter-annotator agreement [9, 66]. Table 6 shows the agreement rate between human annotators and LLMs across different subsets. LLMs achieves an average agreement of 90.4% with human judgments, demonstrating strong alignment with expert evaluations. Notably, the majority of disagreement cases occur in borderline *Somewhat Conflict* instances, suggesting that LLMs is particularly reliable in identifying clear-cut conflict or non-conflict cases. These results support the use of LLMs as practical and effective tools for scalable conflict identification.

## A.7 Self-Consistency Filtering for Reliable Parametric Knowledge Extraction

In this subsection, we assess the effectiveness of our self-consistency-based filtering method in extracting reliable parametric knowledge from LLMs. The core idea is to filter out unstable model beliefs by leveraging generation consistency: for each query, we prompt the model five times and identify the most frequent answer and its occurrence frequency. Queries with low answer frequency likely reflect uncertain or non-committal model behavior, making them unreliable for evaluating the model's true reliance on internal knowledge. To quantify this effect, we group data into sub-datasets based on answer frequency, and apply our "Conflict Detection" method to retain only instances where knowledge conflicts are detected. We then evaluate ConR and MemR on each sub-dataset.

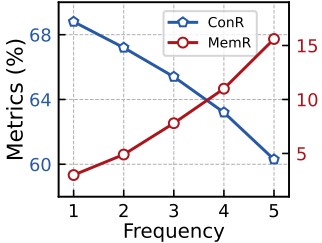

Figure 8: Performance comparison of ConR and MemR across sub-datasets grouped by the answer frequency of LLMs.

As shown in Figure 8, a clear trend emerges: as answer frequency increases, ConR decreases while MemR increases. This suggests that when the model becomes more consistent in its responses, it also tends to rely more heavily on internal (parametric) knowledge, leading to a higher rate of unfaithful generation. Conversely, instances with an answer frequency of 1 exhibit minimal reliance on parametric knowledge (MemR = 3%), indicating that their apparent faithfulness may result from the model's uncertainty rather than true contextual alignment.

These results validate the importance of consistency-based filtering: only when the model confidently expresses its parametric knowledge can we meaningfully assess and intervene in cases of unfaithful generation. This approach also distinguishes our methodology from prior studies [43, 65], which do not account for the stability of model beliefs.

## A.8 Additional Experimental Details

This subsection describes the training prompt, training data, and experimental setup for our study.

**Prompts.** For all methods except Attr_prompt and O&I_prompt, we use a simple QA-format prompt template, following Zhou et al. [79].

> **Base Prompt**
>
> {*context*} Q: {*question*} ? A: {*answer*}.

**Training Datasets.** During the training stage of ParamMute, we construct the training data by randomly sampling 32,580 instances from the combined training sets of the six sub-datasets included in our benchmark, all of which are derived from the MRQA 2019 benchmark [18].

**Experimental Setup.** In this work, all models are trained for 2,100 steps with a total batch size of 32 and a learning rate of 1e-4. To enhance training efficiency, we implement ParamMute with LoRA [24]. For ParamMute, we set the number of suppressed UA-FFNs layers to $N = 8$, and the suppression coefficient in Eq. 11 is fixed at 0.0. The hyperparameters $\alpha$ and $\beta$, which control the relative contributions of $\mathcal{L}_{\text{KAT}}$ and $\mathcal{L}_{\text{KPO}}$ in Eq.8, are both set to 0.5. Additionally, we adopt a dynamic $\gamma$ in $\mathcal{L}_{\text{KPO}}$ (Eq. 10), which linearly transitions from an initial margin ($\gamma_0 = 1$) to a final margin ($\gamma^* = 5$) as training progresses. This adaptive strategy gradually reduces the model's reliance on internal parametric knowledge, encouraging it to rely more on external knowledge. To facilitate faithful evaluation on CoFaithfulQA, we adopt a controlled setting for each dataset–following prior works [1, 29, 58]–to ensure that the provided documents are sufficient to answer the questions, thereby isolating the model's faithfulness from retrieval quality.

## A.9 Implementation Details of Baselines

This subsection describes the implementation details of all baseline methods.

We adopt two prompt-based baselines designed to reflect common prompting strategies: the attributed prompt (Attr_prompt), which directly asks the model to state factual knowledge, and the opinion-and-instruction prompt (O&I_prompt), which combines subjective framing with task-oriented instructions. The corresponding prompt templates are shown below:

> **Attr based prompt**
>
> {*context*} Q: {*question*} based on the given text? A: {*answer*}.

> **O&I based prompt**
>
> Bob said "{*context*}" Q: {*question*} in Bob's opinion? A: {*answer*}.

For the SFT baseline, we incorporate context during training, similar to ParamMute, while keeping the remaining experimental settings identical. To construct preference pairs for DPO training, we use contextually aligned answers from the dataset as "preferred responses" to ensure the consistency with the provided context. The "rejected responses" are generated by identifying parametric knowledge conflicts through our data construction methodology (§4). For KAFT, we employ a hybrid dataset containing both counterfactual and factual data. Specifically, we integrate the counterfactual data developed by Xie et al. [65], leveraging their advanced data construction framework. For DDR, we follow the strategy described in Li et al. [39] to construct preference data. Specifically, for each training instance, we generate multiple outputs under different decoding conditions by varying the sampling temperature and enabling or disabling the use of retrieved context. Each output is evaluated using an accuracy-based reward function. The responses with the highest and lowest reward scores are selected as the positive and negative samples, respectively, for DPO training.

By maintaining an equivalent dataset size and ensuring comparable data quality across all baselines, we provide a rigorous and fair comparison with our proposed ParamMute.

## A.10 Implementation Details of Different Suppression Strategies

This subsection provides implementation details of four suppression strategies designed to reduce the influence of specific model components. These strategies are introduced to investigate how different types of internal suppression affect contextual faithfulness. All methods are applied to the same set of layers identified using the approach in Section 3.1, and implemented on a shared model backbone (LLaMA3-8B-Instruct) to ensure fair comparison. For consistency, we use a uniform suppression coefficient of $\lambda = 0.0$, effectively nullifying the contribution of the targeted submodules.

**FFN Suppression (ParamMute).** We identify a fixed set of unfaithfulness-associated FFN sublayers (as described in Section 2.2) and suppress them by scaling the hidden activations after the nonlinearity with a suppression coefficient $\lambda = 0.0$ in Eq. 11.

**Multi-Head Attention (MHA) Suppression.** For Multi-Head Attention suppression, we target the same transformer blocks selected for the ParamMute setting and suppress the MHA modules within these layers by scaling their output by a factor of $\lambda$.

**Parameter Suppression .** Inspired by SNIP [34], we explored a more fine-grained suppression strategy targeting individual parameters crucial for faithfulness. To achieve this, we followed the SNIP criterion to compute a saliency score for each parameter within the identified FFN layers, defining the score as the product of the parameter's value and the gradient of the loss with respect to that parameter. We then select the top-$k$ parameters with the highest saliency scores, where $k$ is set to match the total number of parameters suppressed in our FFN suppression strategy. These parameters are suppressed by applying a binary mask matrix scaled by the suppression coefficient $\lambda$, effectively modulating their contribution without altering the remaining model weights. This setup aligns the overall suppression magnitude with that of FFN suppression, allowing for a more consistent comparison between strategies.

**Layer Suppression.** We apply suppression to the same set of transformer blocks used in the FFN suppression strategy. For each selected block, we scale the output of the entire block–comprising both the multi-head attention and FFN submodules–by the suppression coefficient $\lambda$ during inference. This allows us to assess the impact of suppressing entire transformer layers while keeping the number and location of suppressed blocks consistent across strategies.

## A.11 Failure Case Analysis

To better understand the task's challenges and our model's limitations, we conducted a failure case analysis. Specifically, following the methodology of Kortukov et al. [32], we manually analyzed 50 incorrect predictions made by ParamMute on the CoFaithfulQA dataset and grouped them into five distinct error types, as detailed in Table 7.

| Error Type | Ratio |
|---|---|
| Partial Match | 48% |
| Annotation Error | 20% |
| Context Ambiguity | 6% |
| Hallucination | 4% |
| Parametric Hallucination | 22% |

Table 7: Distribution of error types.

Among these errors, we identified 11 instances where the model reverted to its own parametric knowledge instead of using the provided context. These cases typically involved factual information such as dates, counts, or specific entities (e.g., people, locations) tied to events. We hypothesize that this occurs when the model has memorized these facts from its training data, leading to overly confident predictions. To investigate this, we generated five outputs for each of the 11 corresponding questions. In 8 of these 11 cases, the model produced the exact same incorrect answer across all five generations, confirming its high confidence. This suggests that while suppressing knowledge pathways is effective, future advancements may require mechanisms that can dynamically arbitrate between internal knowledge and external evidence, especially in cases of high model confidence.

| Models | HotpotQA | | | | Natural-Questions | | | | NewsQA | | | |
|---|---|---|---|---|---|---|---|---|---|---|---|---|
| | ConR ↑ | MemR ↓ | MR ↓ | EM ↑ | ConR ↑ | MemR ↓ | MR ↓ | EM ↑ | ConR ↑ | MemR ↓ | MR ↓ | EM ↑ |
| Vanilla-RAG | 56.90 | 14.51 | 20.31 | 11.63 | 45.64 | 19.23 | 29.65 | 2.48 | 57.34 | 9.14 | 13.75 | 3.61 |
| Attr$_{\text{prompt}}$ | 51.37 | 14.72 | 22.27 | 2.45 | 44.00 | 16.89 | 27.74 | 0.00 | 56.88 | 7.34 | 11.42 | 0.68 |
| O&I$_{\text{prompt}}$ | 42.05 | 11.98 | 22.18 | 1.61 | 41.38 | 10.01 | 19.48 | 0.07 | 48.53 | 5.98 | 10.97 | 0.45 |
| SFT | 69.01 | 7.55 | 9.86 | 64.33 | 58.41 | 10.24 | 14.92 | 57.98 | 64.02 | 5.63 | 8.08 | 53.50 |
| KAFT | 68.75 | 6.87 | 9.08 | 64.75 | 60.89 | 10.50 | 14.71 | 60.11 | 65.35 | 5.37 | 7.59 | 56.21 |
| ParamMute | **71.90** | **6.63** | **8.44** | **65.87** | **61.60** | **9.59** | **13.47** | **60.82** | **67.83** | **4.99** | **6.85** | **57.11** |

| Models | SearchQA | | | | SQuAD | | | | TriviaQA | | | |
|---|---|---|---|---|---|---|---|---|---|---|---|---|
| | ConR ↑ | MemR ↓ | MR ↓ | EM ↑ | ConR ↑ | MemR ↓ | MR ↓ | EM ↑ | ConR ↑ | MemR ↓ | MR ↓ | EM ↑ |
| Vanilla-RAG | 68.56 | 9.92 | 12.64 | 36.64 | 70.52 | 9.30 | 11.66 | 4.81 | 64.93 | 9.91 | 13.24 | 15.38 |
| Attr$_{\text{prompt}}$ | 65.44 | 10.41 | 13.72 | 11.87 | 69.39 | 9.08 | 11.57 | 0.94 | 60.63 | 9.52 | 13.57 | 3.91 |
| O&I$_{\text{prompt}}$ | 53.02 | 9.30 | 14.92 | 2.71 | 65.89 | 6.83 | 9.39 | 0.36 | 55.28 | 8.34 | 13.11 | 0.00 |
| SFT | 78.94 | 6.94 | 8.08 | 76.27 | 80.44 | 4.09 | 4.84 | 70.74 | 61.15 | 8.47 | 12.17 | 55.93 |
| KAFT | 79.04 | 7.56 | 8.73 | 76.47 | 80.27 | 4.18 | 4.95 | 71.78 | 60.23 | 9.13 | 13.16 | 55.80 |
| ParamMute | **80.71** | **6.04** | **6.96** | **77.45** | **81.62** | **4.00** | **4.67** | **71.82** | **62.84** | **6.78** | **9.74** | **57.24** |

Table 8: Performance of different methods on CoFaithfulQA under the noisy retrieval setting.

## A.12 Generalization to Noisy Retrieval Settings

In our main results (§ 6), we follow recent RAG faithfulness literature [1] by adopting a controlled setting where a single retrieved document is guaranteed to contain the answer. This setup is necessary, as faithfulness can only be meaningfully evaluated when the answer is present in the provided context. However, retrieval in real-world applications is typically noisy. To broaden our evaluation and better assess the performance of ParamMute in realistic scenarios, we evaluate our method under a noisy retrieval setting. Specifically, we conducted evaluations on all six CoFaithfulQA sub-datasets (using both $\mathcal{D}^+$ and $\mathcal{D}^-$), following the retrieval protocol from Li et al. [39]. During both training and inference, the model receives the ground-truth passage along with the top-two retrieved passages (deduplicated against the ground-truth), with the order randomly shuffled to prevent positional bias. In addition to our main results, we also report Exact Match (EM) scores.

As shown in Table 8, ParamMute consistently outperforms all baselines across most datasets, achieving the highest faithfulness and the lowest reliance on parametric memory under the noisy retrieval setting. These results further demonstrate the robustness and generalizability of ParamMute in realistic RAG scenarios.

| Model | Vanilla-RAG | Attr$_{prompt}$ | O&I$_{prompt}$ | COIECD | SFT | KAFT | ParamMute |
|---|---|---|---|---|---|---|---|
| LLaMA-3-8B | 63.6 | 64.9 | 64.0 | 64.7 | 63.2 | 65.2 | **67.4** |
| Qwen-2.5-7B | 55.9 | 57.0 | 49.4 | 59.6 | 59.7 | 60.4 | **62.7** |

Table 9: Performance comparison of different methods on FaithEval. Experiments are conducted on both LLaMA3-8B-Instruct and Qwen2.5-7B-Instruct. The best results for each model are highlighted in bold, while the second-highest scores are underlined.

## A.13 Faithfulness Evaluation on FaithEval

While our main evaluation focuses on the proposed CoFaithfulQA, which primarily targets Temporal Misalignment [67] and CoFiQA [1], both categorized as Misinformation Pollution according to the taxonomy of Xu et al. [67], it is also important to assess the generalizability of ParamMute on additional faithfulness benchmarks. To this end, we further evaluate ParamMute on the counterfactual subset of the FaithEval benchmark [67], which can also be classified as Misinformation Pollution under the same taxonomy [67].

The results are shown in Table 9. It can be observed that ParamMute consistently outperforms all baselines on FaithEval. Specifically, ParamMute achieves improvements of 2.4% and 2.3% over the strongest baseline on LLaMA3-8B-Instruct and Qwen2.5-7B-Instruct, respectively. These results further highlight the robustness of ParamMute and demonstrate the effectiveness of suppressing parametric knowledge activation in enhancing model faithfulness.

| Models | GSM8K COT (8) | GPQA (5) | CoQA | Average |
|---|---|---|---|---|
| SFT | **64.06** | 29.24 | 50.92 | 48.07 |
| ParamMute ($\lambda$=0.0) | 9.93 | 27.90 | 45.55 | 27.79 |
| ParamMute ($\lambda$=0.5) | 54.36 | 28.79 | 52.32 | 45.16 |
| ParamMute ($\lambda$=1.0) | 63.70 | **30.58** | **57.5** | **50.59** |

Table 10: Closed-book performance on non-contextual tasks. We use 8-shot Chain-of-Thought (CoT) prompting for GSM8K and 5-shot prompting for GPQA. The Supervised Fine-Tuning (SFT) model serves as the baseline. The notion ParamMute ($\lambda = x$) indicates our model was trained with full suppression ($\lambda = 0$) and then evaluated with a suppression coefficient of $x$ during inference.

## A.14 What is the Impact of UA-FFNs Suppression?

To evaluate whether suppressing the UA-FFNs affects the model's performance, we assess its closed-book performance on a range of tasks under non-contextual settings, including mathematical reasoning and general knowledge probing. Specifically, we use the GSM8K [8], GPQA [54], and CoQA [53] datasets. For fair comparison, all methods are evaluated with the `lm-evaluation-harness` [19], following the protocols described in [28, 41]. We compare the SFT to our proposed ParamMute under different values of $\lambda$ during inference. Notably, although ParamMute is trained with $\lambda = 0$, our soft suppression mechanism (see Eq. 11) enables flexible adjustment of $\lambda$ at inference time.

The results are shown in Table 10. Setting $\lambda = 0$ leads to a substantial drop in performance, with an average decrease of 20.28% compared to the SFT baseline, indicating that our method can effectively suppress the use of parametric knowledge. As $\lambda$ increases, the model's performance gradually recovers and ultimately surpasses the SFT baseline when $\lambda = 1.0$, achieving an average improvement of 2.52%. These findings demonstrate that ParamMute not only preserves performance on non-contextual tasks, but also provides flexible control over the contribution of internal knowledge, allowing the model to adapt to different requirements by adjusting $\lambda$ during inference.

## A.15 How Activation Strength Shapes Parametric Knowledge Reliance?

To better understand how activation strength affects the model's reliance on internal parametric knowledge, we conduct experiments under both the zero-shot and knowledge-adapted settings. Specifically, we evaluate Memory Recall (MemR) and Memorization Ratio (MR) across a range of suppression coefficients $\lambda$ on ConFiQA and CoFaithfulQA.

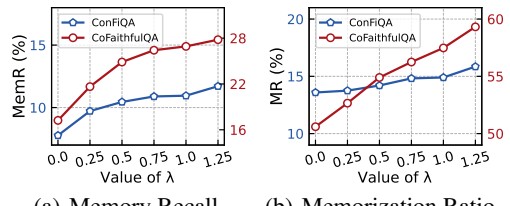

(a) Memory Recall.    (b) Memorization Ratio.

Figure 9: Trends in Memory Recall (MemR) and Memorization Ratio (MR) under varying suppression coefficients $\lambda$, evaluated on ConFiQA and CoFaithfulQA. Each point reflects the average metric across all subsets within the respective benchmark.

As shown in Figure 9, panels (a)–(b) report results for the original model without fine-tuning. In both cases, we observe that decreasing $\lambda$–i.e., applying stronger suppression to UA-FFNs activations–consistently reduces MemR and MR, indicating that suppression effectively reduces reliance on internal parametric memory (lower MemR), without degrading the model's use of external context, as evidenced by the expected decline in MR with decreasing $\lambda$.

These findings empirically highlight the relationship between FFN activation strength and the model's dependency on parametric knowledge. Moreover, they demonstrate the potential of activation-level control as a mechanism for modulating knowledge reliance, offering practical insights for flexibly balancing internal memory and contextual grounding in downstream applications.

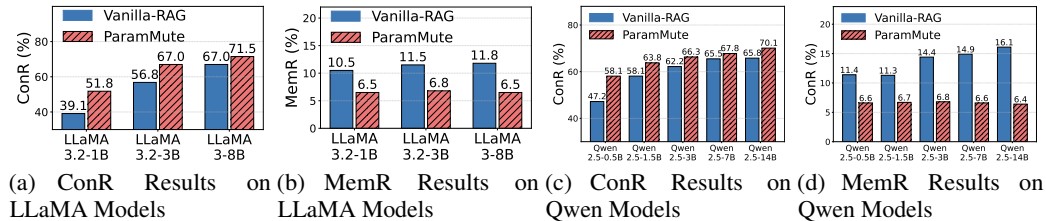

(a) ConR Results on LLaMA Models    (b) MemR Results on LLaMA Models    (c) ConR Results on Qwen Models    (d) MemR Results on Qwen Models

Figure 10: Average ConR and MemR across different models based on the LLaMA and Qwen series, before and after applying ParamMute.

## A.16 Extending ParamMute to More LLMs

We extend ParamMute to a diverse range of LLMs, encompassing multiple model families and sizes. Specifically, our evaluation includes LLaMA3-8B-Instruct, LLaMA3.2-1B-Instruct, LLaMA3.2-3B-Instruct, Qwen2.5-0.5B-Instruct, Qwen2.5-1.5B-Instruct, Qwen2.5-3B-Instruct, Qwen2.5-7B-Instruct, and Qwen2.5-14B-Instruct. The results on ConR and MemR are summarized in Figures 10, while Table 11 presents the average performance of all models on CoFaithfulQA and ConFiQA. This comprehensive evaluation demonstrates the versatility and scalability of ParamMute across a wide spectrum of model architectures and sizes.

These experimental results also illustrate several key insights: 1) Larger models tend to rely more on parametric memory. As model size increases in both the LLaMA and Qwen families, MemR also grows, indicating a tendency to overlook external knowledge in favor of internal parameters. ParamMute counteracts this behavior, decreasing larger models' MemR score to even below that of smaller models. 2) ParamMute consistently benefits all evaluated models. Across both LLaMA

| Models | CoFaithfulQA | | | ConFiQA | | |
|---|---|---|---|---|---|---|
| | ConR ↑ | MemR ↓ | MR ↓ | ConR ↑ | MemR ↓ | MR ↓ |
| LLaMA-3-8B | 63.37 | 10.89 | 14.9 | 22.52 | 31.15 | 59.77 |
| +ParamMute | 69.54 | 6.18 | 8.34 | 69.83 | 8.60 | 11.24 |
| LLaMA-3.1-8B | 59.53 | 10.83 | 15.84 | 15.38 | 29.97 | 68.98 |
| +ParamMute | 68.45 | 6.65 | 9.10 | 71.12 | 9.01 | 11.44 |
| LLaMA-3.2-1B | 35.44 | 9.63 | 21.74 | 32.09 | 18.32 | 36.28 |
| +ParamMute | 49.79 | 6.21 | 11.27 | 62.70 | 7.63 | 11.38 |
| LLaMA-3.2-3B | 53.13 | 10.67 | 17.02 | 26.16 | 23.47 | 49.05 |
| +ParamMute | 65.04 | 6.50 | 9.28 | 69.61 | 8.39 | 11.09 |
| Qwen-2.5-0.5B | 43.55 | 10.50 | 19.39 | 50.72 | 17.15 | 26.20 |
| +ParamMute | 56.17 | 6.33 | 10.34 | 67.54 | 8.04 | 11.03 |
| Qwen-2.5-1.5B | 54.46 | 10.42 | 16.39 | 51.69 | 19.87 | 28.23 |
| +ParamMute | 61.82 | 6.44 | 9.69 | 69.61 | 8.35 | 11.05 |
| Qwen-2.5-3B | 58.60 | 13.59 | 18.79 | 25.47 | 29.34 | 55.70 |
| +ParamMute | 64.35 | 6.45 | 9.31 | 66.30 | 8.62 | 11.94 |
| Qwen-2.5-7B | 62.78 | 13.53 | 17.79 | 24.75 | 33.09 | 59.04 |
| +ParamMute | 65.79 | 6.30 | 8.94 | 69.54 | 8.85 | 11.58 |
| Qwen-2.5-14B | 62.13 | 15.27 | 19.66 | 7.86 | 32.88 | 83.71 |
| +ParamMute | 68.05 | 6.13 | 8.48 | 71.70 | 8.90 | 11.29 |

Table 11: Average performance of LLMs on CoFaithfulQA and ConFiQA before and after applying ParamMute.

and Qwen model families, ParamMute outperforms Vanilla-RAG by boosting accuracy and context faithfulness, underscoring its broad applicability and effectiveness. 3) Not all parameters in RAG models are essential. Pruning parametric knowledge not only reduces computation costs but also fosters better generalization without sacrificing accuracy, highlighting the potential of building a parameter-efficient LLM within the RAG framework.

## B  Limitations and Societal Impacts

**Limitations.** While our method demonstrates consistent improvements across multiple benchmarks, several aspects remain open for future exploration.

Firstly, to facilitate the evaluation of faithfulness in retrieval-augmented generation, CoFaithfulQA is constructed under a controlled setting where the retrieved context is guaranteed to contain sufficient information to answer the question. As a result, unfaithful responses caused by retrieval failures are not reflected in this benchmark. We aim to extend the benchmark to cover a diverse range of task scenarios in future work, thus providing a more comprehensive evaluation of contextual faithfulness in LLMs. Secondly, our intervention strategy focuses on suppressing a specific subset of FFN layers based on activation patterns. While effective, this design operates at a relatively coarse granularity. Exploring finer-grained interventions, such as at the level of individual neurons, may yield further gains in controlling parametric knowledge influence. Finally, due to computational constraints, our experiments are conducted on models of moderate scale. Although our findings generalize across multiple model families, future work could investigate whether similar patterns hold in larger-scale models, and whether scaling effects introduce new challenges or opportunities for intervention.

**Societal Impacts.** Enhancing the faithfulness of retrieval-augmented language models can significantly improve the reliability of AI systems in real-world applications, such as question answering, digital assistants, and knowledge-based services. By reducing the likelihood of generating factually incorrect or misleading responses, our method contributes to safer and more trustworthy deployment of large language models in practice. Furthermore, the proposed activation suppression mechanism offers a flexible means of controlling the model's reliance on parametric knowledge. This flexibility enables task-specific adaptation—dynamically increasing or decreasing dependence on internal memory according to contextual demands—making our approach potentially beneficial in a wide range of downstream scenarios where different levels of grounding are required, such as healthcare, finance, and scientific research, where factual consistency and evidence alignment are particularly critical.

