# OpenReview forum: "ParamMute: Suppressing Knowledge-Critical FFNs for Faithful Retrieval-Augmented Generation"
_NeurIPS.cc/2025/Conference — NeurIPS 2025 poster_

### Official Review · Reviewer_EqkC · 2025-06-27

**Clarity:** 4
**Significance:** 3
**Originality:** 3
**Rating:** 5
**Confidence:** 4

**Summary:**

The authors propose ParamMute, a parametric knowledge muting framework that improves contextual faithfulness by suppressing the activation of unfaithfulness-associated FFNs and calibrating the model toward retrieved knowledge. They observe that mid-to-deep FFN layers (UA-FFNs) become highly activated when generating unfaithful responses. These layers are considered to contain heavy internal knowledge, which leads the model to generate unfaithful outputs.
To address this, the authors introduce ParamMute, which suppresses the activation of such UA-FFN layers and incorporates an adaptation module to encourage reliance on external knowledge. This module includes knowledge-augmented fine-tuning and knowledge preference optimization.
They also introduce a new benchmark, CoFaithfulQA, designed to evaluate faithfulness in cases where there is a conflict between the model's parametric knowledge and external knowledge.
ParamMute demonstrates strong performance on both the CoFaithfulQA and ConFiQA benchmarks, generating more faithful responses.

**Questions:**

I recommend moving the baseline descriptions and the experiments on key hyperparameters from the appendix into the main text.

**Ethical Concerns:**

["Major Concern: Data quality and representativeness"]

**Final Justification:**

My concerns were solved during the discussion, thus I will keep the rating towards acceptance.

**Limitations:**

yes

**Paper Formatting Concerns:**

No issues

**Quality:**

3

**Strengths And Weaknesses:**

**Strengths**
1. The paper is clearly written.
2. The authors present an analysis showing that mid-to-deep FFN layers become highly activated when generating unfaithful responses, and demonstrate that suppressing these layers reduces reliance on parametric knowledge.
3. The authors propose ParamMute, a two-stage framework that reduces activation of UA-FFN layers and incorporates a plug-and-play adaptation module to recalibrate the model's knowledge utilization preferences, and show that it leads to generating more faithful respones.
4. The authors conduct various experiments, including hyperparameter search and exploration of different model choices.

**Weaknesses**
1. The motivation for constructing the CoFaithfulQA benchmark is somewhat unclear. How does it differ from the ConFiQA benchmark? If you select instances where the external evidence is accurate and the parametric knowledge is incorrect, can this task be generalized across various base models, considering that each may have different parametric knowledge?
2. Does the choice and ranking of UA-FFN layers remain consistent across different datasets and base models? Since D- and D+ depend on the dataset and model, it is unclear whether these layers vary with different settings.

---

> ### Author Rebuttal · Authors · 2025-07-30
>
> We sincerely thank the reviewer for the positive evaluation and for the constructive, insightful feedback provided. We greatly appreciate your careful reading of our manuscript and your thoughtful comments, which have helped us to further clarify and strengthen our work. Below, we address each of your concerns in detail.
>
> > Weakness 1: The motivation for constructing the CoFaithfulQA benchmark is unclear. How does it differ from the ConFiQA benchmark? Additionally, if external evidence is accurate but parametric knowledge is incorrect, can this evaluation task be generalized across different base models, given that each model may possess different parametric knowledge.
>
> **Reply to Weakness 1:**
>
> We sincerely appreciate the reviewer's valuable comments and thoughtful questions regarding (1) the differences between our CoFaithfulQA benchmark and ConFiQA, and (2) the generalizability of our task across various base models.
>
> **Motivation and Differences from ConFiQA**:
>
> CoFaithfulQA is designed to evaluate the contextual faithfulness of RAG models in realistic, naturally occurring knowledge conflict scenarios, where the retrieved context is accurate but the model's internal knowledge may be outdated or incorrect (i.e., **temporal misalignment**) [1]. Such situations are common in practical deployments but are underrepresented in existing benchmarks. In contrast, ConFiQA primarily relies on LLM-generated counterfactual contexts to create artificial contradictions for stress-testing, which, according to [2], falls under the category of **Misinformation Pollution**. While both benchmarks aim to advance faithfulness evaluation for LLMs, they target different types of knowledge conflicts and are therefore complementary.
>
> **Generalizability across Different Base Models**:
> We agree with the reviewer's observation that our initial "unfaithful" subset is constructed based on conflicts identified with a specific base model (e.g., LLaMA3-8B), and therefore, some conflict cases may differ across models due to variations in internal knowledge. However, the core challenge of our task-improving faithfulness in situations where incorrect internal knowledge conflicts with correct external evidence-is broadly applicable to all LLMs. More importantly, our contribution extends **beyond a static dataset**: we present a general, reproducible, and user-friendly pipeline for automatically constructing CoFaithfulQA-style benchmarks. This model-agnostic pipeline can be easily used by any researcher to generate conflict cases and evaluate the contextual faithfulness of any model of interest, providing a practical and scalable solution for diverse research needs.
>
> To facilitate adoption and demonstrate the applicability of our approach, we will open source the data generation pipeline, along with example CoFaithfulQA subsets for several mainstream models (e.g., GPT-4o, LLaMA3.2-70B, Qwen2.5-32B, Qwen2.5-7B).
>
> In the revised manuscript, we will further clarify these distinctions and emphasize the general applicability of our proposed evaluation framework. We sincerely thank the reviewer again for highlighting these important points.
>
> > Weakness 2: Does the choice and ranking of UA-FFN layers remain consistent across different datasets and base models? Since D- and D+ depend on the dataset and model, it is unclear whether these layers vary with different settings.
>
> **Reply to Weakness 2:**
>
> We sincerely thank the reviewer for raising this important and insightful question. We agree that the definitions of D− and D+ depend on the specific model and dataset, and that the set of identified UA-FFN layers could, in principle, vary across different settings. To better assess the generalizability of our findings, we conducted two additional experiments: (1) constructing D+ and D− with different datasets, and (2) using different model architectures and sizes to construct D+ and D−. As figures cannot be included in the rebuttal, we report the $D_-(R)$ and $D_+(R)$, as well as $\Delta R^l$, in the table below. Due to space constraints, we have omitted layers with negligible $\Delta R^l$ values, and only a subset of representative UA-FFN layers are shown (with some layers within each range skipped).
>
>
> ||0|1|2|18|19|21|23|25|27|29|31|
> |------|--------|--------|--------|--------|--------|--------|--------|--------|--------|--------|--------|
> |$\mathbb{E}_{\mathcal{D}-}[R^l(\hat{r})]$|0.2693|0.3584|0.3114|0.1502|0.1593|0.1617|0.1866|0.1879|0.1852|0.2585|0.6179|
> |$\mathbb{E}_{\mathcal{D}+}[R^l(\hat{r})]$|0.2818|0.3613|0.3097|0.1466|0.1466|0.139|0.1589|0.1565|0.1559|0.2362|0.6118|
> |$\Delta R^l$|-0.126|-0.0288|0.0175|**0.0365**|**0.1266**|**0.2266**|**0.2768**|**0.314**|**0.2926**|**0.2233**|0.0609|
>
> Table 1: Differences in Activation Patterns of LLaMA3-8B-Instruct (32 layers) on HotpotQA. Notably, $\Delta R^l$ values are substantially larger between layers 19 and 29.
>
>
> ||0|1|2|18|19|21|23|25|27|29|31|
> |------|--------|--------|--------|--------|--------|--------|--------|--------|--------|--------|---------|
> |$\mathbb{E}_{\mathcal{D}-}[R^l(\hat{r})]$|0.2734|0.3642|0.3151|0.1512|0.1608|0.1585|0.1743|0.1626|0.156|0.2276|0.5982|
> |$\mathbb{E}_{\mathcal{D}+}[R^l(\hat{r})]$|0.2828|0.3645|0.3114|0.1455|0.1439|0.1296|0.1402|0.1275|0.1251|0.206|0.602|
> |$\Delta R^l$|-0.0931|-0.0026|0.0373|0.0577|**0.1697**|**0.2888**|**0.3409**|**0.3519**|**0.3085**|**0.2161**|-0.0376|
>
> Table 2: Differences in Activation Patterns of LLaMA3-8B-Instruct (32 layers) on TriviaQA. Notably, $\Delta R^l$ values are substantially larger between layers 19 and 29.
>
> ||0|1|2|3|42|43|44|45|47|49|51|53|55|62|63|
> |-|-|-|-|-|-|-|-|-|-|-|-|-|-|-|-|
> |$\mathbb{E}_{\mathcal{D}-}[R^l(\hat{r})]$|0.3116|0.0007|0.0012|0.005|0.2413|0.2471|0.2878|0.2612|0.2088|0.199|0.1626|0.1658|0.1775|0.334|0.315|
> |$\mathbb{E}_{\mathcal{D}+}[R^l(\hat{r})]$|0.3212|0.0008|0.0012|0.0053|0.2389|0.2415|0.2746|0.2428|0.1972|0.1869|0.1552|0.1593|0.1731|0.3381|0.3182|
> |$\Delta R^l$|-0.0961|0|-0.0022|-0.0026|0.0236|0.0562|**0.1323**|**0.1836**|**0.1166**|**0.12**|**0.0741**|**0.0651**|0.0436|-0.0409|-0.032|
>
> Table 3: Differences in Activation Patterns of Qwen2.5-32B-Instruct  (64 layers) on CoFaithfulQA. Notably, $\Delta R^l$ values are substantially larger between layers 44 and 53.
>
> ||0 |1 |47|48|51|54|57|60|63|66|69|70|78|79 |
> |------|--------|--------|--------|--------|--------|--------|--------|--------|--------|--------|--------|--------|--------|---------|
> |$\mathbb{E}_{\mathcal{D}-}[R^l(\hat{r})]$|0.4653|0.3861|0.3715|0.3841|0.3792|0.3523|0.349|0.332|0.3137|0.3102|0.2946|0.2868|0.5594|0.7412|
> |$\mathbb{E}_{\mathcal{D}+}[R^l(\hat{r})]$|0.4692|0.3817|0.3507|0.3611|0.3621|0.3373|0.3346|0.3192|0.2993|0.3011|0.2923|0.2866|0.5556|0.7425|
> |$\Delta R^l$|-0.0395|0.0442|**0.2082**|**0.2294**|**0.1702**|**0.1503**|**0.1431**|**0.1276**|**0.1438**|**0.0909**|0.0235|0.0018|0.0376|-0.0126|
>
> Table 4: Differences in Activation Patterns of LLaMA3-70B-Instruct (80 layers) on CoFaithfulQA. Notably, $\Delta R^l$ values are substantially larger between layers 47 and 66.
>
> The results consistently demonstrate that **our key finding is robust**: excessive activation in a specific subset of mid-to-deep FFN layers (typically 60%–85% of the model depth) is strongly associated with unfaithful outputs, regardless of dataset, model family, or size. This not only highlights the generality of our observation but also provides valuable insights into the mechanisms underlying unfaithful generation and the role of parametric knowledge in LLMs. We hope these findings will inform future research. In the revised version, we will more clearly highlight the stability and broad applicability of this pattern. We thank the reviewer again for their valuable feedback.
>
> In summary, we thank the reviewer for your high evaluation and for your valuable suggestions. Your feedback has enabled us to improve the clarity, depth, and rigor of our analysis, and we look forward to incorporating these improvements in the revised manuscript. We hope that our additional clarifications and new results address your concerns and further demonstrate the contribution and generalizability of our work.
>
> ---
> [1] Kortukov E, et al. Studying large language model behaviors under context-memory conflicts with real documents. COLM2024.
>
> [2] Xu R, et al. Knowledge Conflicts for LLMs: A Survey. EMNLP2024.

---

> > ### Comment · Reviewer_EqkC · 2025-08-01
> >
> > Thank you for your detailed analysis and explanation. I will keep my positive score.

---

> ### Author Response · Authors · 2025-08-01
>
> Dear Reviewer EqkC,
>
> Thank you very much for your recognition of our work and the positive evaluation. We greatly appreciate your thoughtful engagement with our paper and are pleased that the new results helped to clarify and strengthen our contributions. All newly added experiments will be incorporated into the revised version. Once again, we sincerely thank you for your constructive feedback.
>
> Best regards,
>
> The Authors

---

### Official Review · Reviewer_MDE9 · 2025-06-28

**Clarity:** 3
**Significance:** 3
**Originality:** 3
**Rating:** 4
**Confidence:** 2

**Summary:**

This paper presents ParamMute, a novel framework aimed at improving the contextual faithfulness of large language models (LLMs). It tackles the challenge of LLMs overly relying on internal parametric knowledge instead of retrieved evidence. ParamMute reduces this bias by selectively suppressing feed-forward networks (FFNs) that are highly correlated with unfaithful generations. To further reinforce grounding in external content, it incorporates a plug-and-play adaptation module that strengthens the model’s reliance on retrieved information. It also introduces CoFaithfulQA, a benchmark derived from six diverse QA datasets, designed to evaluate faithfulness in the presence of conflicting knowledge. Extensive experiments on CoFaithfulQA and ConFiQA show that ParamMute significantly improves faithfulness while reducing dependence on internal knowledge.

**Questions:**

1. Can the authors clarify whether the performance gains shown in Table 2 are statistically significant compared to the baselines?

2. Could the authors report the accuracy of the methods in Table 2? This would help readers better understand how well the models are performing on the core task beyond faithfulness metrics.

3. Can the authors include or summarize qualitative examples of failure cases? Such analysis would offer valuable insight into the types of errors the model still makes and the limitations of the proposed approach.

**Ethical Concerns:**

["NO or VERY MINOR ethics concerns only"]

**Final Justification:**

After reading the rebuttal, my concerns have been addressed. Thus, I keep my positive score for this paper.

**Limitations:**

yes

**Paper Formatting Concerns:**

No concerns

**Quality:**

3

**Strengths And Weaknesses:**

This paper has the following strengths:

1. It proposes a novel method, Parametric Knowledge Muting through FFN Suppression (ParamMute), which improves contextual faithfulness by suppressing FFNs associated with unfaithful generation and calibrating the model toward retrieved evidence.

2. It introduces CoFaithfulQA, a new benchmark tailored to evaluate faithfulness in settings where internal knowledge conflicts with accurate external sources, offering a valuable resource for future research.

3. Extensive experiments demonstrate that ParamMute consistently improves faithfulness across both CoFaithfulQA and the existing ConFiQA benchmark, substantially reducing reliance on parametric memory.

However, this paper has the following weaknesses:

1. The results in Table 2 do not clearly show that ParamMute significantly outperforms baseline methods; the improvements appear marginal.

2. Reporting the accuracy metric in Table 2 would help assess each method’s effectiveness on the underlying QA tasks.

3. Including qualitative analysis of failure cases would provide deeper insights into the challenges of the task and the model’s limitations.

---

> ### Author Rebuttal · Authors · 2025-07-30
>
> We sincerely thank Reviewer MDE9 for the helpful and insightful comments. We appreciate the time and effort you devoted to reviewing our work and for raising several important points that have helped us improve both the clarity and rigor of our manuscript. Below, we address each of your concerns in detail.
>
> > Weakness 1 & Question1: The improvements in Table 2 appear marginal. Can the authors clarify whether the observed performance gains are statistically significant compared to the baselines?
>
> **Reply to Weakness 1 & Question1:**
>
> We appreciate the reviewer's emphasis on this important aspect. We agree that conducting statistical significance tests to evaluate performance gains will further strengthen the rigor of our results. Following this suggestion, we have performed significance testing against all baselines on both ConR and MemR, as MR is a derived metric based on ConR and MemR and therefore cannot be directly subjected to significance testing. The results are shown in the table1 and table2 below. The findings show that the improvements of ParamMute are statistically significant across most benchmarks and baselines. In the revised manuscript, we will present these statistical analyses alongside the main results to enhance the clarity and rigor of our performance comparisons.
>
> ||HotpotQA ConR↑|HotpotQA MemR↓|HotpotQA EM↑|NQ ConR↑|NQ MemR↓|NQ EM↑|NewsQA ConR↑|NewsQA MemR↓|NewsQA EM↑|
> |-|-|-|-|-|-|-|-|-|-|
> |vanilla-rag|60.34†|13.88†|9.35†|53.09†|14.41†|2.84†|60.27†|8.24†|5.03†|
> |Attr_prompt|58.93†|13.95†|3.46†|55.36†|11.07†|0.41†|58.87†|7.56†|1.61†|
> |O & I|47.79†|10.72†|1.83†|49.25†|8.23|0.61†|52.03†|5.3†|1.5†|
> |COIECD|62.51†|12.19†|24.73†|56.21†|12.28†|12.47†|61.44†|6.31†|33.83†|
> |SFT|70.92|6.24|63.69†|59.76†|10.29†|55.31†|61.97†|5.08†|54.39†|
> |KAFT|69.52†|6.87†|63.48†|60.89|9.23|58.96|65.09|4.74†|53.32†|
> |C-DPO|67.21†|7.64†|51.56†|62.24|9.79†|40.09†|61.47†|4.96†|36.42†|
> |DDR|68.66†|7.15†|41.16†|63.29|10.33†|0†|64.74†|4.85|37.6†|
> |ParamMute|71.06|6.17|66.06†|60.68|9.08|58.76|65.24|4.85|55.57|
>
> Table 1: Statistical significance analysis on HotpotQA, NQ, and NewsQA. A † symbol indicates that ParamMute outperforms the corresponding baseline with statistical significance ($p < 0.05$). Exact match (EM) scores are also reported to provide further insight into QA task performance.
>
> | |SearchQA ConR↑|SearchQA MemR↓|SearchQA EM↑|SQuAD ConR↑|SQuAD MemR↓|SQuAD EM↑|TriviaQA ConR↑|TriviaQA MemR↓|TriviaQA EM↑|
> |-|-|-|-|-|-|-|-|-|-|
> |vanilla-rag|66.76†|10.55†|23.45†|77.93†|6.79†|8.8†|61.8|11.47†|10.23†|
> |Attr_prompt|62.53†|10.55†|5.28†|77.35†|6.38†|3.02†|59.97†|10.43†|9.59†|
> |O & I|52.26†|9.23†|4.68†|76.81†|6.11†|2.55†|55.41†|8.1†|1.14†|
> |COIECD|69.74†|11.66†|20.91†|73.12†|7.64†|9.02†|63.62|11.99†|19.95†|
> |SFT|75.29†|6.87†|69.27†|79.17†|4.28†|68.31†|61.75†|8.34†|41.97†|
> |KAFT|77.38†|7.43†|73.15†|80.04†|4.18|73.5†|62.32|8.41†|47.41†|
> |C-DPO|76.21†|5.62|56.11†|80.08†|5.26†|50.81†|58.62†|8.47†|34.93†|
> |DDR|78.07†|6.48†|0†|81.36†|4.36†|41.36†|60.71†|7.73†|27.93†|
> |ParamMute|78.76|6.04|75.89|80.58|4.04|72.18|61.9|6.91|56.44|
>
> Table 2: Statistical significance analysis on SearchQA, SQuAD, and TriviaQA. A † symbol indicates that ParamMute outperforms the corresponding baseline with statistical significance ($p < 0.05$).
>
> > Weakness 2 & Question2 : Could the authors report the accuracy metric in Table 2 to better assess each method's effectiveness on the core QA task beyond faithfulness?
>
> **Reply to Weakness 2 & Question 2:**
>
> We thank the reviewer for this helpful suggestion. We would like to clarify that the **Contextual Recall (ConR) metric reported in Table 2 is equivalent to the standard accuracy** metric commonly used in QA evaluation. ConR is defined as the percentage of questions for which the model's answer is faithful to the context, determined by whether the answer exactly matches the gold answer. To further address the reviewer's concern regarding model performance on QA tasks, we have **also reported exact match (EM) scores**; the results are shown in the table provided in response to Weakness 1 & Question 1. ParamMute consistently outperforms all baselines on both accuracy and EM.
>
> We will make this correspondence explicit in the revised manuscript to ensure that readers clearly understand the relationship between ConR and standard QA metrics, and we will also include EM results in the revised version to further clarify the effectiveness of ParamMute on QA tasks.
>
>
> > Weakness 3 & Question 3: Including qualitative examples or analysis of failure cases would provide valuable insights into the types of errors the model makes and the limitations of the proposed approach. Can the authors provide such examples?
>
> **Reply to Weakness 3 & Question 3:**
>
> We thank the reviewer for this valuable suggestion. We agree that qualitative analysis of failure cases can provide deeper insights into the limitations of our approach and the challenges of the task. In response, we performed an error analysis on 50 incorrect predictions and categorized the failure cases as follows:
>
> - Type1: Model is correct but not exactly equal to the ground truth: 24 cases
> - Type2: Incorrect ground truth: 6 cases
> - Type3: Ground truth is ambiguous or too long: 4 cases
> - Type4: Context is unclear: 3 cases
> - Type5: Model's answer is not derived from the provided context: 2 cases
> - Type6: Model persists with incorrect parametric knowledge: 11 cases
>   - Numerical issues: 7 cases
>     - Year: 4 cases
>     - Counting: 3 cases
>   - Person or location related to a specific event: 3 cases
>
> Among all cases where the model produced incorrect answers, there are 11 instances in which the model persisted with its own parametric knowledge. Most of these involve numerical information-such as years and counts-or specific people or locations related to particular events. We suspect this may be due to the model having encountered these samples during training, resulting in highly confident predictions. To investigate this, we sampled each question five times for these cases and found that in 8 out of 11 instances, the model produced the same answer across all 5 outputs. This indicates that, although ParamMute substantially alleviates reliance on parametric knowledge through FFN suppression, this issue still persists. We provide several representative failure cases in the table below. For most of these cases, we do not include the full context due to space limitations, especially when the error type is unrelated to the context or when the context is excessively long.
>
>
> | Query| Error Type| Answers| Pred. | Context|
> |----------------------------------------------------------------------------------------------------------|------------|----------------------------------------------------------------------------------------------------|------------------------|-|
> | Name a musician that worked with Hiren Roy and Vilayat Khan.| Type1| Nikhil Banerjee| Ravi Shankar| Hiren Roy  Hiren Roy (1920–1992) was for many years considered to be the best sitar maker in India.  Many great musicians, including **Nikhil Banerjee**, Vilayat Khan, Annapurna Devi, and **Ravi Shankar**, have sworn by his creations. \n  Vilayat Khan  Ustad Vilayat Khan (28 August 1928 – 13 March 2004) was one of India's well known sitar maestros.  Along with **Ravi Shankar**, Ali Akbar Khan, **Nikhil Banerjee** and his younger brother Imrat Khan, Vilayat Khan helped introduce Indian Classical Music to the West.|
> | who holds the record for most platinum albums| Type2| AC / DC| The Beatles| \ |
> | where did the last name keith come from| Type3| a nickname , derived from the Middle High German Kit , a word meaning` sprout , offspring | East Lothian, Scotland| \ |
> | Rafter, framing & T-bevel are useful types of this "geometric" tool| Type5| square| a saw| \ |
> | when is the last time the dolphins went to the superbowl| Type6| 1984| 1973| \ |
> | Which agency is involved in negotiations with pirates holding Capt.Phillips?| Type6| FBI| U.S. Navy| \ |
>
> Table 3: Representative Examples of Common Failure Cases
>
> We will further expand this qualitative analysis in the revised manuscript to provide readers with a more comprehensive understanding of the challenges and potential directions for future improvement.
>
> We once again thank Reviewer MDE9 for the constructive feedback and thoughtful suggestions. Your comments have enabled us to strengthen our experimental analysis and clarify key aspects of our work. We will incorporate all the additional analyses, clarifications, and experimental results discussed above in the revised manuscript to provide a more comprehensive and transparent presentation.

---

> > ### Comment · Reviewer_MDE9 · 2025-08-01
> >
> > I appreciate your detailed analysis and thoughtful explanation. I've decided to keep my positive score.

---

> ### Author Response · Authors · 2025-08-01
>
> Dear Reviewer MDE9,
>
> We sincerely appreciate your thoughtful feedback and the time you dedicated to helping us improve our work. Your positive score is truly encouraging, and we are pleased that our efforts have addressed your concerns. We will incorporate the additional experiments into the revised version to further strengthen the contribution of our paper. Once again, thank you for your valuable comments and support.
>
> Best regards,
>
> The Authors

---

### Official Review · Reviewer_4WNi · 2025-07-02

**Clarity:** 3
**Significance:** 3
**Originality:** 3
**Rating:** 4
**Confidence:** 3

**Summary:**

This paper introduces ParamMute, a framework designed to enhance the faithfulness of RAG in LLMs. The authors observe that unfaithful generation—where model outputs contradict retrieved evidence—often stems from over-reliance on internal parametric knowledge embedded in specific feed-forward network (FFN) layers. ParamMute mitigates this issue by identifying and suppressing Unfaithfulness-Associated FFNs (UA-FFNs) to reduce parametric interference and by incorporating a plug-and-play adaptation module that calibrates the model to prioritize external evidence.

**Questions:**

1. Are the findings (i.e., "when a specific subset of mid-to-deep FFN layers exhibits excessive activation, the model tends to rely more heavily on its internal knowledge, consequently producing unfaithful outputs") generalizable to models with larger sizes (e.g., 70B)?
2. Are there any patterns in the selected set of FFN layers, such as distribution between shallow or deep layers, and does layer selection have any connections to model size?
3. During evaluation, is the selected set of layers fixed across different tasks?
4. How would suppressing FFNs impact the model's performance in non-retrieval situations (i.e., simply removing all retrieved documents during the evaluation)?
5. Regarding the retrieval setup, based on Appendix A.7, it appears that retrieval recall is intentionally set as 100% for all benchmarks. Can the model be evaluated in a more realistic setup with imperfect retrieval? Would the proposed method still achieve high faithfulness?

**Ethical Concerns:**

["NO or VERY MINOR ethics concerns only"]

**Final Justification:**

I've read the authors' responses and other reviewers' comments, and I remain positive about the work.

**Quality:**

3

**Strengths And Weaknesses:**

### Strengths
- The paper provides a compelling explanation for unfaithful generation, linking it to specific FFN layers. Causal intervention experiments (e.g., increased NLL loss when suppressing UA-FFNs) robustly support this claim.

- ParamMute is lightweight (suppression + adaptation) and effective across diverse model families (LLaMA, Qwen) and sizes.

- The introduced CoFaithfulQA benchmark is useful for future research, and the proposed method is validated on both CoFaithfulQA and ConFiQA, demonstrating promising potential.

### Weaknesses

- My main concern is about the limited scope of the evaluation: the paper assumes that the retrieved guideline fragments are always sufficient and accurate, but does not test robustness under noisy or incomplete retrieval, which would better reflect real-world conditions.
- Reducing reliance on parametric knowledge may undermine the model's factuality in non-RAG scenarios.
- It is unclear how sensitive the model is to the suppression coefficient. Plotting a curve with standard errors to study its impact would provide valuable insights.

---

> ### Author Rebuttal · Authors · 2025-07-30
>
> We sincerely thank Reviewer 4WNi for the constructive and insightful feedback. We appreciate your thoughtful questions and suggestions, which have helped us further clarify and improve our work. Below, we address each of your concerns in detail.
>
> > W1 & Q5: The current evaluation assumes that the retrieved guideline fragments are always sufficient and accurate, and does not assess robustness under noisy or incomplete retrieval, which would be more representative of real-world scenarios. Can the model still maintain high faithfulness in such a realistic setup with imperfect retrieval?
>
> **Reply to W1 & Q5:**
>
> We thank the reviewers for raising this important point regarding the evaluation scope and retrieval setup. We fully agree that real-world retrieval is often noisy, and that evaluating robustness under such conditions is crucial for practical deployment. In our current submission, we adopt a controlled setting-consistent with recent RAG faithfulness literature [1-2]-mainly because (1) faithfulness evaluation requires that the provided context contains all key information, and (2) introducing noisy passages would add confounding variables, such as inter-context contradictions, complicating the analysis of faithfulness mechanisms [3].
>
> Nevertheless, we agree that evaluating ParamMute in more realistic RAG scenarios-including noisy retrieval-is important for demonstrating its generalizability and robustness. To this end, we conducted additional experiments on all six CoFaithfulQA sub-datasets (using both D+ and D-) following the retrieval protocol described in [4]: during both training and inference, the model receives the ground-truth passage and the top-2 retrieved passages (deduplicated against the ground-truth), with order randomly shuffled. We compare ParamMute with three prompting baselines, SFT, and KAFT; results are shown in the table1 below.
>
> ||AVG. ConR↑|AVG. MemR↓|HotpotQA ConR↑|HotpotQA MemR↓|NQ ConR↑|NQ MemR↓|NewsQA ConR↑|NewsQA MemR↓|SearchQA ConR↑|SearchQA MemR↓|SQuAD ConR↑|SQuAD MemR↓|TriviaQA ConR↑|TriviaQA MemR↓|
> |-|-|-|-|-|-|-|-|-|-|-|-|-|-|-|
> |Vanilla-rag|60.65|12.00|56.90|14.51|45.64|19.23|57.34|9.14|68.56|9.92|70.52|9.30|64.93|9.91|
> |Attr_prompt|57.95|11.33|51.37|14.72|44.00|16.89|56.88|7.34|65.44|9.13|69.39|9.08|60.63|9.52|
> |O&I_prompt|51.03|8.74|42.05|11.98|41.38|10.11|48.53|9.58|53.02|9.30|65.89|6.83|55.28|8.47|
> |SFT|68.66|7.15|69.01|7.55|58.41|10.24|65.02|8.14|78.94|6.94|80.44|4.07|65.11|8.34|
> |KAFT|69.09|7.27|68.75|6.87|60.89|10.50|65.35|8.57|79.04|7.56|80.27|4.18|62.84|6.94|
> |ParamMute|**71.08**|**6.34**|**71.90**|**6.63**|**61.60**|**9.59**|**67.28**|**6.85**|**80.71**|**6.04**|**81.62**|**4.00**|**68.24**|**6.78**|
>
> Table 1: Comparison of Methods in Noisy RAG Settings
>
> ParamMute **consistently outperforms baselines across most datasets**, achieving the highest faithfulness and lowest reliance on parametric memory, even with noisy retrieval. These results demonstrate its robustness and generalizability in realistic RAG scenarios. We will include these findings to provide a more comprehensive evaluation under imperfect retrieval.
>
> > W2 & Q4: Reducing reliance on parametric knowledge may undermine the model's factuality in non-RAG (closed-book) scenarios. How does suppressing FFNs affect the model's performance when no retrieved documents are provided during evaluation?
>
> **Reply to W2 & Q4:**
>
> We thank the reviewer for highlighting the importance of evaluating ParamMute in non-RAG (closed-book) scenarios. We agree that reducing reliance on parametric knowledge may undermine the model's factuality in such cases, and we have confirmed this in our supplementary experiments, as shown by the ParamMute ($\lambda = 0$) results in the table2 below. This outcome is expected, as suppressing the model's dependence on parametric knowledge inevitably reduces its ability to recall facts from internal memory in the absence of external evidence.
>
> However, we would like to emphasize that ParamMute is designed to **provide fine-grained control over parametric knowledge** (see Eq. 7), allowing users to dynamically adjust $\lambda$ during inference to balance internal knowledge and external evidence as needed. To demonstrate this flexibility, we also evaluated ParamMute with $\lambda$ set to 0.5 and 1.0 during inference. As shown in the table2, gradually increasing $\lambda$ enables the model to recover factuality, even outperforming the SFT baseline when $\lambda = 1$. We will include these experiments in the revised version to further demonstrate the **adaptability and practical** utility of ParamMute across diverse scenarios.
>
> ||HotpotQA|NQ|NewsQA|SearchQA|SQuAD|TriviaQA|AVG.|
> |-|-|-|-|-|-|-|-|
> |SFT|**15.5**|**13.83**|5.76|62.17|13.12|52.14|27.09|
> |ParamMute($\lambda$=0)|4.25|3.10|1.97|14.13|3.61|12.59|6.61|
> |ParamMute($\lambda$=0.5)|12.45|10.91|5.07|54.6|10.89|43.61|22.92|
> |ParamMute($\lambda$=1)|15.22|13.24|**5.97**|**62.92**|**13.30**|**52.68**|**27.22**|
>
> Table2: Results on CoFaithfulQA in non-retrieval (closed-book) settings. The reported metric is accuracy.
>
> > W3:It is unclear how sensitive the model is to the suppression coefficient. Plotting a curve with standard errors to study its impact would provide valuable insights.
>
> **Reply to W3:**
>
> We fully agree that plotting a curve to study the model's sensitivity to the suppression coefficient is important. As shown in Appendix A.9, we have already examined the effects of key hyperparameters, including the suppression coefficient **(Figure 5(a))**. To further address this point, we reran the experiments with different values of $\lambda$ three times each and report the mean and standard error in the table3 below, as figures are not allowed in the rebuttal. The results confirm the same trend as in the paper: **increasing $\lambda$ strengthens the model's reliance on parametric knowledge and reduces contextual faithfulness**. We will update the figure in the revised version to include standard errors for a more comprehensive presentation.
>
> |$\lambda$|0.0|0.25|0.5|0.75|1.0|
> |-|-|-|-|-|-|
> |ConR↑ Mean|69.54|69.44|69.27|69.23|69.14|
> |ConR↑ Mean ± SE|69.54 ± 0.058|69.44 ± 0.047|69.27 ± 0.078|69.23 ± 0.072|69.14 ± 0.11|
> |MemR↓ Mean|6.18|6.33|6.63|6.94|7.23|
> |MemR↓ Mean ± SE|6.18 ± 0.07|6.33 ± 0.049|6.63 ± 0.13|6.94 ± 0.052|7.23 ± 0.055|
>
> Table 3: Performance Metrics (Mean ± SE) for Different Suppression Coefficients
>
> > Q1 & Q2:Are the findings about the relationship between mid-to-deep FFN activation and unfaithful outputs generalizable to larger models (e.g., 70B)? Additionally, are there any patterns in the selected set of FFN layers-such as their distribution across shallow or deep layers-and does layer selection depend on model size?
>
> **Reply to Q1 & Q2:**
>
> We thank the reviewer for raising these important questions about the generalizability of our findings to larger models and the selection patterns of FFN layers. To address this, we conducted additional experiments on **Qwen2.5-32B and LLaMA3-70B**. We report $D_-(R)$, $D_+(R)$, and $\Delta R^l$ in the table4 below. Due to space constraints, only a subset of representative UA-FFN layers is shown.
>
> ||0|1|47|48|51|54|57|60|63|66|69|70|78|79|
> |-|-|-|-|-|-|-|-|-|-|-|-|-|-|-|
> |$\mathbb{E}_{\mathcal{D}-}[R^l(\hat{r})]$|0.4653|0.3861|0.3715|0.3841|0.3792|0.3523|0.349|0.332|0.3137|0.3102|0.2946|0.2868|0.5594|0.7412|
> |$\mathbb{E}_{\mathcal{D}+}[R^l(\hat{r})]$|0.4692|0.3817|0.3507|0.3611|0.3621|0.3373|0.3346|0.3192|0.2993|0.3011|0.2923|0.2866|0.5556|0.7425|
> |$\Delta R^l$|-0.0395|0.0442|**0.2082**|**0.2294**|**0.1702**|**0.1503**|**0.1431**|**0.1276**|**0.1438**|**0.0909**|0.0235|0.0018|0.0376|-0.0126|
>
> Table4: Differences in Activation Patterns of LLaMA3-70B-Instruct (80 layers) on CoFaithfulQA. Notably, $\Delta R^l$ values are substantially larger between layers **47 and 66**. For Qwen2.5-32B results, see Table 3 in our response to Reviewer EqKC.
>
> The results confirm the **robustness of our key finding**: excessive activation in a specific subset of mid-to-deep FFN layers (typically 60%–85% of the depth) is consistently associated with unfaithful outputs, even for the largest models tested. Notably, the distribution of these layers is similar across different model sizes and architectures, indicating this pattern is not model-specific. We will include detailed results and visualizations in the revised manuscript. These findings suggest that our conclusions generalize well and provide valuable insight into the mechanism underlying unfaithful generation in LLMs, which we hope will inform and inspire future research. We will include the detailed results and visualizations in the revised manuscript.
>
> > Q3:During evaluation, is the selected set of layers fixed across different tasks?
>
> **Reply to Q3:**
>
> Yes, the selected set of layers for a given model is fixed across different tasks and datasets, and is determined based on the model's activation gap on the D+ and D- subsets. Once identified, these layers remain unchanged during evaluation. We will clarify this procedure further in the revised manuscript to strengthen the overall presentation.
>
> We sincerely thank Reviewer 4WNi again for the thoughtful feedback and constructive suggestions. We believe the newly added analyses and experiments address your concerns and further strengthen the empirical foundation and clarity of our work. We hope our responses and revisions will improve the paper's quality and make its contributions more compelling.
>
> ---
> [1] Bi B, et al. Context-dpo: Aligning language models for context-faithfulness[J]. ACL2025.
>
> [2] Huang L, et al. Improving contextual faithfulness of large language models via retrieval heads-induced optimization[J]. ACL2025.
>
> [3] Xu R, et al. Knowledge Conflicts for LLMs: A Survey. EMNLP2024.
>
> [4] Li X, et al. Rag-ddr: Optimizing retrieval-augmented generation using differentiable data rewards[J]. ICLR2025.

---

> ### Author Response · Authors · 2025-08-05
>
> Dear Reviewer 4WNi,
>
> Thank you very much for your thoughtful, comprehensive, and encouraging review. We sincerely appreciate your constructive suggestions, which contributed meaningfully to the improvements made during the rebuttal phase. These refinements have helped further enhance the quality and clarity of our work. We hope our responses effectively address your concerns and are helpful for your final assessment. If there are any remaining concerns, please feel free to let us know. We are happy to further clarify or discuss any points as needed. Once again, thank you for your detailed and insightful feedback.
>
>
>
> Best regards,
>
> The Authors

---

> > ### Comment · Reviewer_4WNi · 2025-08-05
> >
> > Thank you for the detailed responses. I've also read other reviewers' comments, and I remain positive about the work.

---

> > > ### Author Response · Authors · 2025-08-06
> > >
> > > Dear Reviewer 4WNi,
> > >
> > > Thank you very much for your positive feedback and for taking the time to review our work as well as consider the comments from other reviewers. We truly appreciate your support and thoughtful assessment throughout the entire rebuttal process. We will incorporate all new experiments and discussions into the revised version to further strengthen the contribution of our paper. Once again, thank you for your valuable comments and support.
> > >
> > > Best regards,
> > >
> > > The Authors

---

### Official Review · Reviewer_iu2z · 2025-07-03

**Clarity:** 3
**Significance:** 3
**Originality:** 3
**Rating:** 4
**Confidence:** 5

**Summary:**

The paper hypothesizes that the faithfulness of the LLM in RAG setting is influenced by unfaithfulness-related FFN activation, which causes the model to use internal parametric knowledge and hallucination. Then the paper propose a way to suppress such action to improve contextual faithfulness to prioritize towards external contextual knowledge in RAG settings. This include identify and suppress the targeted activations, and then fine-tune with SFT and KTO on contextual QA dataset. The paper create a new benchmark called CoFaithfulQA and show their method improve the faithfulness significantly

**Questions:**

NA

**Ethical Concerns:**

["NO or VERY MINOR ethics concerns only"]

**Final Justification:**

The authors made significant effort in rectifying the paper with supplemental experiments. The paper is more sound.

**Limitations:**

yes

**Quality:**

3

**Strengths And Weaknesses:**

## Strength
- The paper is novel in analyzing unfaithfulness by disecting the activation of FFN layers for unfaithful responses and identify those activation as source of unfaithfulness. It provide correlation analysis on when and where those neurons are activated when an unfaithful response is generated.
- The method of suppressing the activation based on the faithful and unfaithful subsets is simple. The training part, which is SFT and KTO, is straightforward and intuitive.
- Experiments are conducted throughly and extensively.


## Weakness
- The new benchmark is highly duplicative of FaithEval (https://arxiv.org/abs/2410.03727). This is not mentioned or compared in the paper. The authors need to add this to the paper.
- While the unfaithful neurons identification is novel, the training part is not novel, but intuitive.
- **A big question remained unanswered**: What is actually lost after the deactivation of those FFNs? The paper should investigate more, by analyzing closed-book performance on various non-contextual tasks, such as math reasoning, GPQA. How much raw parametric knowledge have been lost (or gained?) after such process? Are there any unintended consequences of such pruning process?

---

> ### Author Rebuttal · Authors · 2025-07-30
>
> We sincerely thank Reviewer iu2z for the thoughtful and constructive feedback, and for recognizing the strengths of our work. Below, we first address the primary concern regarding the assessment of non-contextual tasks after deactivating the Unfaithfulness-Associated FFNs (UA-FFNs), and then provide point-by-point responses to the remaining comments.
>
> > Weakness 3: What is actually lost after deactivating those FFNs? The paper should further investigate by analyzing closed-book performance on non-contextual tasks (e.g., math reasoning, GPQA). How much raw parametric knowledge is lost (or gained) after this process, and are there any unintended consequences of such pruning?
>
> **Reply to Weakness 3:**
>
> We appreciate the reviewer's suggestion to analyze the effects of suppressing UA-FFNs on non-contextual tasks. In our original submission, we did not include experiments on closed-book or general knowledge tasks, as our primary focus was to **improve the contextual faithfulness of LLMs in RAG scenarios** by mitigating their reliance on internal knowledge.
>
> That said, we agree that evaluating the impact of UA-FFN suppression on non-contextual tasks can further clarify the boundaries of our method and help readers understand its scope. In response to the reviewer's suggestion, we have conducted additional experiments on several non-contextual tasks, including GSM8K, GPQA, and CoQA. We used the lm-evaluation-harness and followed the evaluation protocol in [3] for fair comparison. We compared four methods: the SFT baseline, and our proposed ParamMute with different settings of the suppression coefficient $\lambda$ at inference time. Notably, although ParamMute is trained with $\lambda$=0, our soft suppression mechanism (as described in Eq. 7) allows $\lambda$ to be flexibly adjusted during inference, enabling fine-grained control over the contribution of internal parametric knowledge. The results are summarized in the table1 below:
>
> | Model         | GSM8K COT(8) | GPQA(5) | CoQA | AVG   |
> |-|-|-|-|-|
> | SFT  | **64.06**        | 29.24  | 50.92 | 48.07 |
> | ParamMute($\lambda$=0)  | 9.93  | 27.90    | 45.55 | 27.79 |
> | ParamMute($\lambda$=0.5) | 54.36 | 28.79   | 52.32 | 45.16 |
> | ParamMute($\lambda$=1)   | 63.70  | **30.58**   | **57.5**  | **50.59** |
>
> Table 1: Comparison of Different Methods on Non-Contextual Tasks
>
> As the results show, setting $\lambda$ to 0 leads to a significant reduction in factual recall on non-retrieval tasks, confirming the intended effect of reducing the model's dependence on parametric knowledge. Importantly, this reduction in factual recall can be fully recovered by increasing $\lambda$ (e.g., setting $\lambda=1$); in non-RAG scenarios, the model's factual recall even surpasses the SFT baseline when $\lambda=1$. This demonstrates that ParamMute provides practitioners with the flexibility to adjust $\lambda$ according to their needs, **without unintended consequences, thereby further broadening its practical applicability**. Once again, we thank the reviewer for their insightful and constructive feedback. We believe these analyses significantly strengthen the empirical foundation of our work and demonstrate the practical flexibility of ParamMute in both RAG and non-RAG scenarios. We will incorporate these findings and explicitly highlight their implications in the revised manuscript.
>
> > Weakness 1: The authors need to mention or compare their work with FaithEval in the paper.
>
> **Reply to Weakness 1:**
> We thank the reviewer for drawing our attention to FaithEval[1]. We would like to respectfully clarify that, although both CoFaithfulQA and the counterfactual subset of FaithEval are designed to evaluate LLM faithfulness under context-memory conflict, our benchmark differs from FaithEval in several important aspects:
>
> **Task Focus**: According to the taxonomy in [2], FaithEval targets **Misinformation Pollution** scenarios, where the retrieved context is intentionally erroneous (e.g., LLM-generated fake evidence), and the model is expected to override its internal knowledge to follow misleading context. In contrast, CoFaithfulQA focuses on **Temporal Misalignment**, where the context is accurate but the model's parametric knowledge is outdated or incorrect. This setup reflects a more prevalent failure case in practical RAG deployments [3], as the model may ignore correct evidence and instead rely on obsolete internal knowledge.
>
> **Construction Pipeline**: FaithEval uses LLM-generated modified contexts, validated by both LLMs and human annotators. In contrast, CoFaithfulQA identifies cases where the model's closed-book answers conflict with the ground-truth context, confirming these instances through multiple LLMs and manual review to ensure genuine disagreement between outdated internal knowledge and accurate external evidence.
>
> Therefore, while both benchmarks contribute to advancing faithfulness evaluation for LLMs, they target different failure modes. To better situate our work within the broader landscape and clarify these distinctions, we fully agree that a more comprehensive discussion of related work-including FaithEval-will strengthen readers' understanding of the field and highlight the unique contributions of CoFaithfulQA. Accordingly, we will **add a dedicated section on this topic** in the revised manuscript.
>
> > Weakness 2: While the unfaithful neurons identification is novel, the training part is not novel, but intuitive.
>
> **Reply to Weakness 2:**
>
> We thank the reviewer for recognizing the novelty of our core contribution-unfaithful neuron (UA-FFN) identification-and for their feedback regarding the training component. We would like to clarify that the adaptation module is not intended as a standalone novelty, but rather as an indispensable second stage that **synergistically complements our novel suppression mechanism**. The suppression stage addresses over-reliance on parametric knowledge by targeting UA-FFNs, while the adaptation stage recalibrates the model to effectively utilize external context. This recalibration is achieved through a **redesigned loss function that combines margin-based preference objectives with SFT principles**, specifically tailored for the RAG contextual faithfulness setting.
>
> Importantly, our experiments (see Tables 2, 3, and 4) demonstrate that the integrated suppression-and-adaptation framework outperforms baseline methods and ablated variants, validating the necessity and effectiveness of combining both stages. Thus, the novelty of ParamMute lies not in individual components, but in the overall design and empirical validation of the complete suppression-and-adaptation architecture. We will revise our manuscript to better clarify this design rationale and how the two stages work together.
>
> In summary, we sincerely thank Reviewer iu2z for the thoughtful feedback. We appreciate your valuable input and will incorporate all additional analyses and clarifications in the revised manuscript. We hope these revisions address your concerns and further strengthen our work.
>
> ---
> [1] Ming Y, et al.FaithEval: Can Your Language Model Stay Faithful to Context, Even If" The Moon is Made of Marshmallows". ICLR2025.
>
> [2] Xu R, et al. Knowledge Conflicts for LLMs: A Survey. EMNLP2024.
>
> [3] Kortukov E, et al. Studying large language model behaviors under context-memory conflicts with real documents. COLM2024.

---

> > ### Comment · Reviewer_iu2z · 2025-08-06
> > **Reviewer response:**
> >
> > About weakness 1:
> > I don't see a functional difference between FaitEval and CoFaitfulEval, regardless of how the context is "different" from pretrained knowledge, the task remains as "you must respect context and ignore internal knowledge". So I believe an eval on this benchmark is warranted.
> >
> > Thanks for the lambda experiment. I will increase score once a complete comparison is done.

---

> ### Author Response · Authors · 2025-08-05
>
> Dear Reviewer iu2z,
>
> I hope this message finds you well. As the discussion period is nearing its end with **less than three days remaining**, I wanted to ensure we have addressed all your concerns satisfactorily. If there are any additional points or feedback you'd like us to consider, please feel free to let us know. Your insights are invaluable to us, and we're eager to address any remaining issues to improve our work.
>
> Thank you for your time and effort in reviewing our paper.

---

> ### Author Response · Authors · 2025-08-06
>
> Dear Reviewer,
>
> We sincerely thank you for the continued discussion and fully agree that both FaithEval and CoFaithfulQA share the same evaluation protocol: the model is expected to "respect context and ignore internal knowledge" when the two are in conflict. We appreciate this suggestion and agree that evaluating our method and baselines on FaithEval is important for completeness and a comprehensive comparison.
>
> In response to your suggestion, we have conducted comprehensive experiments on the counterfactual subset of the FaithEval benchmark. For both LLaMA3-8B-Instruct and Qwen2.5-7B-Instruct, we report the performance of several baselines as well as our proposed ParamMute, as shown in the table below:
>
>
> | Model         | Vanilla-RAG | Attr_prompt | O&I_prompt | COIECD | SFT | KAFT | ParamMute |
> |---------------|-------------|------|-------|--------|---------|-|-----------|
> | LLaMA3-8B |    63.6     | 64.9 |  64.0   | 64.7   |  63.2  | 65.2 | **67.4**  |
> | Qwen2.5-7B |    55.9     | 57.0 | 49.4  |   59.6    |    59.7    |    60.4     | **62.7**|
>
>
> The results demonstrate that ParamMute outperforms all baselines on FaithEval as well, further highlighting its robustness and the effectiveness of suppressing parametric knowledge activation to enhance model faithfulness.
>
> These **new results will be included** in the revised manuscript, accompanied by **a more detailed discussion in Section 4** regarding the distinctions and connections between the two benchmarks. We believe this will clarify their complementary nature and further improve the clarity and overall quality of the paper.
>
> We sincerely hope that our responses have addressed your concerns and will be helpful in your final assessment. Once again, thank you for your detailed and insightful feedback.
>
> Best regards,
>
> The Authors

---

> ### Author Response · Authors · 2025-08-07
>
> Dear Reviewer,
>
> I hope this message finds you well. Thank you again for your valuable feedback and suggestions. As recommended, we have conducted additional experiments on the FaithEval benchmark and will include the new results in our revision, together with a more detailed discussion of the distinctions and connections between FaithEval and CoFaithfulQA. We sincerely hope these updates address your concerns and further strengthen the paper. If you have any remaining questions or require further clarification, please feel free to let us know. Once again, thank you very much for your thoughtful review and valuable feedback.
>
> Best regards,
>
> The Authors

---

> > ### Comment · Reviewer_iu2z · 2025-08-09
> > **Reviewer comment**
> >
> > Thanks for new results. I'm up on the score.

---

> ### Author Response · Authors · 2025-08-09
>
> Dear Reviewer,
>
> Thank you very much for your positive feedback and for considering an increase in the score. We truly appreciate your time and thoughtful review. All the additional experiments and related discussions will be incorporated into the revised version to further strengthen the paper.
>
> Best regards,
>
> The Authors

---

### Note · Authors · 2025-08-12

Dear Reviewers and AC,

We sincerely thank you for your constructive feedback, active engagement, and positive evaluations, as well as for highlighting the strengths of our work:
- Novelty in our main contribution: analyzing unfaithfulness by dissecting the activation of FFN layers for unfaithful responses (Reviewers iu2z, 4WNi, MDE9, EqKC).
- Comprehensive and sufficient experiments (Reviewers iu2z, MDE9, EqKC).
- Clear and well-structured writing (Reviewer EqKC).

**All concerns and suggestions were addressed, acknowledged, and accepted by the reviewers during the rebuttal process**. Among these, two key concerns were: the potential negative impact of suppressing parametric knowledge on non-RAG scenarios, and the representativeness of CoFaithfulQA relative to other counterfactual-based faithfulness benchmarks.

In the revised version, we will incorporate all updates made during the rebuttal phase to further strengthen our work:
- Inference-time λ control experiments: Include in §6 the rebuttal experiments showing that adjusting λ during inference enables flexible control over the involvement of parametric knowledge.
- Benchmark discussion and motivation clarification: In §4, add a more detailed comparison with other counterfactual-based faithfulness benchmarks to clarify our motivation, and append the rebuttal results on FaithEval.
- Generality of core findings: In §6, add rebuttal experiments across different model families and sizes to confirm the generality of our finding that UA-FFNs consistently occur in mid-to-deep layers.
- Clarifications and Additional Analyses: Provide clearer explanations of UA-FFN selection criteria and the relationship between accuracy (Acc) and ConR, along with significance testing results. In the appendix, add experiments evaluating our method under noisy RAG scenarios, together with a discussion of representative failure cases.


Finally, we greatly **appreciate the positive evaluations from all reviewers and their efforts to help us improve our work**. We are pleased to note that our framework offers a novel approach to enhancing contextual faithfulness by suppressing activations of unfaithfulness-associated FFNs and guiding the model toward retrieved knowledge. Furthermore, our investigation into UA-FFNs provides valuable insights that can deepen the community’s understanding of the model’s internal mechanisms.

Best regards,

The Authors

---

### Decision · Program_Chairs · 2025-09-17

**Decision:**

Accept (poster)

**Comment:**

The paper shows that unfaithful generation in retrieval-augmented models can be traced to the excessive activation of a subset of feed-forward networks (FFNs). Based on this insight, the authors propose a novel framework called ParamMute, which improves the contextual faithfulness of language models by identifying and suppressing these unfaithfulness-associated FFNs. The framework also includes an adaptation module to better calibrate the model toward the retrieved external knowledge. To validate their approach, the authors introduce a new benchmark, CoFaithfulQA, specifically designed to evaluate model faithfulness in scenarios where a model's internal knowledge conflicts with accurate external evidence.

On the positive side, the reviewers found that the paper's primary contribution was its novel analysis of unfaithfulness, which dissects the internal mechanisms of the model by linking unfaithful responses to the activation of specific FFN layers. The reviewers also highlighted that the proposed ParamMute method is simple and effective, and they were impressed by the comprehensive and thorough experiments conducted across various models and benchmarks.

The reviewers identified areas of improvement, such as an initial lack of comparison to a similar existing benchmark (FaithEval) and the need to evaluate the potential negative impact of the method on the model's performance in non-retrieval scenarios. Reviewers also suggested testing the framework's robustness in more realistic settings with noisy or imperfect retrieval and requested a more in-depth analysis of the method's generalizability to larger models, its statistical significance, and its common failure cases. During the rebuttal period, the authors addressed all of these points with additional experiments and analyses. By incorporating this valuable reviewer feedback, the final manuscript will be more thorough, polished, and of higher quality.

Based on these reviews, I recommend accepting this paper. The reviewers were excited about the paper's novel approach to improving model faithfulness, noting that the investigation into unfaithfulness-associated FFNs provides valuable insights that can deepen the community’s understanding of the internal mechanisms of large language models. The authors were commended for their active engagement and for thoroughly addressing all reviewer concerns during the discussion period. I expect the authors to incorporate the feedback and the extensive additional experiments conducted during the rebuttal into the final version of the paper (especially generalization and failure case analysis).